# Epigraph-based Multigrid PINNs for Two-player General-sum Differential Games

## Abstract

In continuous-time, noncooperative, safety-critical settings, players with distinct goals and shared state constraints are naturally modeled by two-player general-sum differential games with state constraints. Solving such games requires numerically computing the Nash equilibrium of coupled Hamilton–Jacobi (HJ) PDEs, but these computations suffer from the curse of dimensionality (CoD). Physics-informed neural networks (PINNs) offer a scalable alternative, yet the resulting control is derived by maximizing the Hamiltonian with respect to gradients of the learned value function. Consequently, inaccurate value approximations can result in unsafe closed-loop control policies. Prior attempts to improve the accuracy of learned values have relied on supervised data, but these data are costly to obtain and might be unavailable in high-dimensional settings. To overcome these challenges, we propose Epigraph-based Multigrid PINNs (EMP), a fully self-supervised framework that eliminates dependence on supervised data. EMP introduces a learned value-driven rollout sampling strategy that leverages informative values and their gradients along closed-loop trajectories for PINN training, and applies multigrid refinement to improve the accuracy of value approximations. Together, these components yield safe and scalable control policies. Evaluations on 5D and 9D vehicle systems and a 13D drone system demonstrate that EMP achieves lower collision rates than all epigraph-based baselines under comparable budgets, highlighting its effectiveness for safety-critical multi-agent interactions without supervised data.

## 1 Introduction

General-sum differential games with state constraints become prevalent in modeling safety-critical and noncooperative two-player interactions in continuous time, such as collision avoidance (Fridovich-Keil et al., 2020b; Schwarting et al., 2019; Cleac'h et al., 2019). In these settings, players pursue distinct objectives while operating under shared safety constraints, leading to coupled decision-making dynamics. Computing a Nash equilibrium in this setting requires solving a system of Hamilton–Jacobi (HJ) PDEs, where each player is associated with a distinct value function (Başar & Zaccour, 2018). This is fundamentally more difficult than the zero-sum differential games, which need a single HJ PDE (Berkovitz, 1975). The interdependence across multiple PDEs in general-sum differential games makes both analysis and computation substantially more challenging.

While classical methods can solve HJ PDEs, their computational complexity grows exponentially with state dimensions, which makes them intractable for high-dimensional systems due to the curse of dimensionality (CoD) (Mitchell & Tomlin, 2003). Physics-informed neural networks (PINNs) have emerged as a scalable alternative that alleviates the CoD and approximates PDE solutions (Weinan et al., 2021). However, safety-critical HJ problems remain challenging: state constraints (e.g., when constraints are violated or treated as collision penalties) introduce nonsmooth or even discontinuous value functions, which undermine the approximation accuracy of PINNs (Zhang et al., 2024). In HJ formulations, control policies are derived by maximizing the Hamiltonian that depends on the value gradients. As a result, inaccurate value approximations, especially imprecise gradients, can lead to unsafe control policies. Prior work has shown that accurate value approximations require not only reducing value approximation error but also accurately capturing value gradients (Yu et al., 2022).

To improve the accuracy of value approximations, several recent approaches have improved PINN training with supervised data (Gopakumar et al., 2023; Raissi et al., 2019). However, such approaches

introduce additional limitations since they rely on offline numerical solvers to generate supervision. For example, Pontryagin Maximum Principle (PMP) generates supervised data by solving boundary value problems (BVPs). However, BVP solvers become computationally expensive in high-dimensional systems, often suffer from convergence issues due to their high sensitivity to initial trajectory guesses (Kierzenka & Shampine, 2001), and are prone to failure when singular arcs arise in the optimal solution (Cristiani & Martinon, 2010).

Motivated by these limitations, we propose a fully self-supervised epigraph-based learning framework that eliminates reliance on supervised data. Our method combines value-driven rollout sampling, which provides implicit supervision on both values and gradients, with epigraph-based PINNs for stable value approximations. To further suppress accumulated rollout errors, we extend multigrid refinement via the Full Approximation Scheme (FAS) (Henson et al., 2003), enabling more accurate value approximations for deriving control policies. Together, these components yield reliable value approximations and safe control policies.

We claim our contributions as follows: **(1)** We introduce a learned value-driven rollout sampling strategy together with multigrid refinement for epigraph-based PINN, which significantly improves the accuracy of value approximations and thus leads to the safety of the resulting control policies. **(2)** We provide a comprehensive evaluation on three case studies across both linear and nonlinear dynamics, comparing our method against existing epigraph-based learning approaches. **(3)** Under comparable computational budgets and training dataset size, we show that our framework consistently achieves higher safety performance than all baselines across all case studies.

## 2 RELATED WORK

**Epigraph technique in games.** The epigraph technique is first applied to state-constrained optimal control problems (SC-OCPs), where an auxiliary state variable is introduced to transform discontinuous value functions into continuous ones. Building on this idea, recent studies have leveraged machine learning methods such as Proximal Policy Optimization (PPO) (So & Fan, 2023) and physics-informed machine learning (PIML) (Tayal et al., 2025) to solve SC-OCPs in various single-agent settings. The epigraph technique has also been extended to zero-sum differential games with state constraints, enabling the computation of equilibrium values through numerical solvers (Lee & Tomlin, 2021; Gammoudi & Zidani, 2023). Recent work (Zhang et al., 2024) extends this approach to two-player general-sum differential games with state constraints, proving the existence of viscosity solutions to the corresponding HJ equations and deriving safe control policies through a hybrid method that incorporates epigraph-based PINN with supervision from PMP. However, to the best of our knowledge, the method of utilizing the epigraph-based learning framework to derive safe control policies in two-player differential games without any supervised data remains unexplored.

**HJ equations and PINN.** A common framework for solving differential games is the HJ equation, a first-order nonlinear parabolic PDE. Conventional numerical methods such as level set methods (Osher et al., 2004) and OptimizedDP (Bui et al., 2022) suffer from the CoD because their grid-based discretization scales exponentially with state dimension. Neural network–based PDE solvers, particularly PINN, can alleviate this issue by replacing grids with mesh-free sampling and function approximations (Weinan et al., 2021). Variants of PINNs further mitigate the CoD in solving HJB equations (Hu et al., 2024; Cho et al., 2022), embedding PDE residuals and boundary conditions into the training loss to optimize neural networks. Recent HJ-based learning methods investigate specific formulations, including initial-time value approximations (Han et al., 2018), supervised learning driven by PMP (Nakamura-Zimmerer et al., 2021), and HJ reachability learning (Bansal & Tomlin, 2021). Convergence of PINNs has also been demonstrated when approximating smooth value functions that are the solutions to HJ PDEs (Shin et al., 2020; Ito et al., 2021). However, solving HJ PDEs with state constraints to derive safe control policies through purely self-supervised physics-informed learning remains an open challenge (Lee & Tomlin, 2023).

## 3 METHODOLOGY

In this section, we start to reformulate general-sum differential games with state constraints using the epigraph technique, which yields the corresponding HJ PDEs. Since learning value functions solely from PDEs and boundary conditions often leads to inaccurate approximations, we introduce a gradient

dynamics module into PINN training, enabling value-driven rollouts that provide self-supervised signals for both values and their gradients. To further improve the value approximations for safe control, we adopt a two-level FAS that transfers corrections between coarse and fine time grids, reducing residual errors and refining value approximations. Finally, we present how these components are integrated into the EMP framework. A schematic overview of our method is shown in Fig.1.

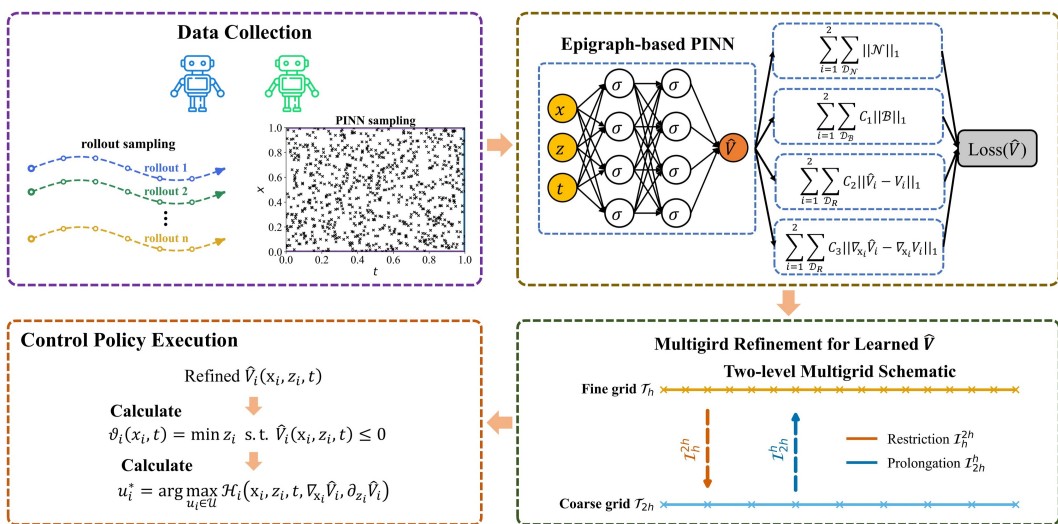

Figure 1: Overview of the EMP algorithm. Training data are generated using two sampling strategies. The value functions $\hat{V}_i$ ($i = 1, 2$) are first trained with the epigraph-based PINN, followed by multigrid refinement. The resulting $\hat{V}_i$ is then used to compute the optimal $z_i$ and derive the corresponding optimal control policies $u_i^*$.

### 3.1 PROBLEM STATEMENT

**Notations and assumptions.** We consider a two-player general-sum differential game with state constraints and complete information. Each player $i \in \{1, 2\}$ follows a time-invariant dynamical system $\dot{x}_i = f_i(x_i, u_i)$ and $x_i(t_0) = \bar{x}_i$, where $x_i \in \mathcal{X}_i \subseteq \mathbb{R}^{d_x}$ is state and $u_i \in \mathcal{U}_i \subseteq \mathbb{R}^{d_u}$ is control input. We use the shorthand $\mathbf{a}_i = (a_i, a_{-i})$ to denote a pair of elements for Player $i$ and its fellow player. The joint state space is $\mathcal{X} = \mathcal{X}_1 \times \mathcal{X}_2$. Player $i$ has an instantaneous loss $l_i : \mathcal{X} \times \mathcal{U} \to \mathbb{R}$ and a terminal loss $g_i : \mathcal{X}_i \to \mathbb{R}$. The feasible set for Player $i$ is $\mathcal{F} = \{\mathbf{x}_i \in \mathcal{X} \mid c_i(\mathbf{x}_i) \leq 0\}$, where $c_i$ is a differentiable function representing the worst-case state constraint violation over possibly multiple constraints. If $c_i(\mathbf{x}_i) > 0$, at least one constraint is violated. We assume in this paper: $\mathcal{U}_i$ is compact and convex; $f_i$ and $c_i$ are Lipschitz continuous; $l_i$ and $g_i$ are Lipschitz continuous and bounded.

Given optimal control $u_{-i}^*$ for Player $-i$, the Nash equilibrium value $\vartheta_i(\mathbf{x}_i, t)$ of Player $i$ is defined as

$$\vartheta_i(\mathbf{x}_i, t) := \min_{u_i \in \mathcal{U}_i} \int_t^T l_i\left(\mathbf{x}_i(s), u_i(s), u_{-i}^*(s)\right) ds + g_i(x_i(T)) \tag{1}$$
$$\text{s.t.} \quad c_i(\mathbf{x}_i(s)) \leq 0, s \in [t, T].$$

Here, the objective of Player $i$ is to minimize its payoffs while satisfying $c_i(\mathbf{x}_i) \leq 0$. The constraints ensure system safety by requiring Player $i$'s trajectory to remain in the feasible set $\mathcal{F}$ over $[t, T]$.

**Epigraph technique and HJ equations.** To compute $\vartheta_i$, we introduce an auxiliary state $z_i \in \mathcal{Z}_i \subseteq \mathbb{R}^{d_z}$ and define an augmented value function $V_i : \mathcal{X} \times \mathcal{Z}_i \times [0, T] \to \mathbb{R}$ using epigraph technique (Altarovici et al., 2013):

$$V_i(\mathbf{x}_i, z_i, t) := \min_{u_i \in \mathcal{U}_i} \max \left\{ \max_{s \in [t, T]} c_i(\mathbf{x}_i(s)), \int_t^T l_i\left(\mathbf{x}_i, u_i, u_{-i}^*\right) ds + g_i - z_i \right\}. \tag{2}$$

The auxiliary state $z_i$ evolves according to $\dot{z}_i = -l_i$ and $z_i(t_0) = \bar{z}_i$. Here, $\bar{z}_i$ represents the value $\vartheta_i$ at a given $(\bar{\mathbf{x}}_i, t_0)$ and is determined as follows: find $\bar{z}_i \in [z_{\min}, z_{\max}]$ such that $V_i(\bar{\mathbf{x}}_i, \bar{z}_i, t_0) = 0$; if $V_i(\bar{\mathbf{x}}_i, z, t_0) > 0$ for all $z \in [z_{\min}, z_{\max}]$, then $\bar{z}_i = +\infty$. The relationship between $V_i$ and $\vartheta_i$ follows

$$\vartheta_i(\mathbf{x}_i, t) = \min z_i, \quad \text{s.t.} \quad V_i(\mathbf{x}_i, z_i, t) \leq 0. \tag{3}$$

Eq. 2 and 3 imply that when $V_i(\mathbf{x}_i, z_i, t) \leq 0$, the state constraints $c_i(\mathbf{x}_i(s)) \leq 0$ hold for all $s \in [t, T]$. In other words, Player $i$ remains safe throughout the entire interaction. In addition, for all $(\mathbf{x}_i, z_i, t) \in \mathcal{X} \times \mathcal{Z}_i \times [0, T]$ and sufficiently small $h > 0$ with $t + h \leq T$, the augmented value function satisfies the following dynamic programming principle:

$$V_i(\mathbf{x}_i, z_i, t) = \min_{u_i \in \mathcal{U}_i} \max \left\{ \max_{s \in [t, t+h]} c_i(\mathbf{x}_i(s)), V_i\left(\mathbf{x}_i(t+h), z_i(t+h), t+h\right) \right\}, \quad (4)$$

Finally, the function $V_i$ is characterized as the viscosity solutions to HJ equations (Zhang et al., 2024) denoted by $(\mathcal{N})$, and satisfies the boundary conditions denoted by $(\mathcal{B})$.

$$\mathcal{N}(V_i, \mathbf{x}_i, z_i, t) := \max \left\{ c_i(\mathbf{x}_i) - V_i(\mathbf{x}_i, z_i, t), \partial_t V_i - \max_{u_i \in \mathcal{U}_i} \mathcal{H}_i(\mathbf{x}_i, z_i, t, \nabla_{\mathbf{x}_i} V_i, \partial_{z_i} V_i) \right\}, \quad (5)$$

$$\mathcal{B}(V_i, \mathbf{x}_i, z_i) := V_i(\mathbf{x}_i, z_i, T) - \max \left\{ c_i(\mathbf{x}_i), g_i(T) - z_i(T) \right\}, \text{ for } i = 1, 2. \quad (6)$$

where $\mathcal{H}_i$ is an augmented Hamiltonian satisfying $\mathcal{H}_i = -\nabla_{\mathbf{x}_i} V_i \cdot \boldsymbol{f}_i + \partial_{z_i} V_i \cdot l_i$. The optimal control for Player $i$ is thus $u_i^* = \arg\max_{u_i \in \mathcal{U}_i} \mathcal{H}_i$ and note that $\mathcal{N}$ for Player $i$ depends on the optimal control $u_{-i}^*$ of Player $-i$.

## 3.2 EPIGRAPH-BASED VALUE GRADIENT DYNAMICS

To address the inaccuracy of values learned purely from PDEs and boundary conditions, we introduce a value gradient dynamics module into PINN's training. Existing HJ-based PINN variants that rely on BVP solvers under PMP take advantage of supervised signals for values and their gradients, yet they come with significant drawbacks. In BVP solvers, the state trajectory $\mathbf{x}_i(t)$ must be generated forward in time, while the value gradient trajectory $\nabla_{\mathbf{x}_i} V_i(t)$ must be generated backward from the terminal condition $\nabla_{\mathbf{x}_i} V_i(T)$. Since these two processes must be matched simultaneously, the simultaneous generation of the two makes convergence highly sensitive to initialization (Nakamura-Zimmerer et al., 2021). In practice, such initial guesses are rarely available, which makes PMP-driven methods highly limited in realistic settings.

To overcome these limitations, we incorporate the dynamics of the value gradient into PINN training. Instead of requiring an external BVP solver and an accurate initial guess, our module generates state trajectories forward in time by applying approximate optimal controls, obtained through Hamiltonian maximization with the current learned value functions. Along these trajectories, both values and their gradients are computed through the corresponding dynamic equations, producing informative trajectories that serve as self-supervised signals. These signals iteratively refine the learned value functions and their gradients toward the ground truth during training (Wang et al., 2025). As a result, the value gradient dynamics module preserves the benefits of informative value and gradient signals while eliminating the dependence on fragile initialization and external supervised data.

To establish the theoretical basis of this module, we follow Theorem 3.4 in Bokanowski et al. (2021) and Theorem 2 in Hermosilla & Zidani (2023) to derive the value gradient dynamics for the game settings in this paper. Defining the augmented value gradients as $q_i^x(t) = \nabla_{\mathbf{x}_i} V_i$, we have

$$\dot{q}_i^x = -q_i^x \cdot \nabla_{\mathbf{x}_i} \boldsymbol{f}_i - \mu_i \nabla_{\mathbf{x}_i} c_i - \lambda_i \nabla_{\mathbf{x}_i} l_i, \quad q_i^x(T) = \lambda_i \nabla_{x_i} g_i. \quad (7)$$

where $\mu_i$ is a non-negative Borel measure on $[0, T]$ that serves as the Lagrange multiplier associated with the state constraints $c_i(\mathbf{x}_i)$, and $\lambda_i \in [0, 1]$ is a scalar multiplier. In our problem settings, we take $\lambda_i = 1$. When state constraints are active at time $t$, meaning $c_i(\mathbf{x}_i(t)) = 0$, $\mu_i(\{t\})$ is an arbitrary positive scalar. Conversely, if the state constraints are never active on $[0, T]$, meaning $c_i(\mathbf{x}_i(t)) < 0$ for all $t$, then $\mu_i([0, T]) = 0$. The full derivation of Eq. 7, along with its explanation and the justification for selecting $(\lambda_i, \mu_i)$, are provided in Appendix A.1.

## 3.3 MULTIGRID APPROACH

While value gradient dynamics provide informative value and gradient signals, they also introduce potential limitations. In particular, value-driven rollouts rely on the current learned value function to generate closed-loop trajectories. Any approximation error in the value function propagates forward along these trajectories and accumulates over time, which can degrade the accuracy and safety of control policies, especially in complex and high-dimensional systems. The failure of such value-driven rollouts for PINNs has been observed in high-dimensional case studies (see Case 3).

To mitigate error accumulation in value-driven rollouts, we extend the FAS (Henson et al., 2003; Ghimire et al., 2025; Riccietti et al., 2022), a multigrid solver for nonlinear PDEs, to refine the value $\hat{V}_i$ learned via epigraph-based PINN. The key idea is to improve value approximations across consecutive time steps $t$ and $t + h$ by enforcing the dynamic programming principle in Eq. 4 through a coarse-to-fine correction process. Specifically, residual errors from the fine grid are projected onto the coarse grid, where they are cheaper to resolve, and the resulting corrections are interpolated back to the fine grid. These recursive corrections suppress the stepwise propagation of errors, leading to more accurate value approximations and safer control policies. The refinement process has four steps: (1) restriction: transfer the fine-grid approximations and residuals to the coarse grid; (2) coarse-grid computation: solve the coarse problem with the restricted residuals; (3) coarse-grid correction: compute the corrections between the restricted fine-grid approximations and the coarse-grid solution; (4) prolongation: interpolate the corrections back to the fine grid to refine the fine-grid approximations. Formally, we denote $\hat{V}_{i,t}^h$ as the learned value for Player $i$ at time $t$ on fine grid $\mathcal{T}_h$ with size $h$. The restriction operator $\mathcal{I}_h^{2h}$ maps fine-grid values $\hat{V}_{i,t}^h$ to the coarse grid $\mathcal{T}_{2h}$ with size $2h$:

$$\mathcal{I}_h^{2h}(\hat{V}_{i,t}^h) = \begin{cases} 0.25\hat{V}_{i,t-h}^h + 0.5\hat{V}_{i,t}^h + 0.25\hat{V}_{i,t+h}^h, & \text{if } t = nh, n \in \mathbb{N}^+ \\ 0.5(\hat{V}_{i,t}^h + \hat{V}_{i,t+h}^h), & \text{if } t = 0 \end{cases}, \quad (8)$$

Similarly, the prolongation operator $\mathcal{I}_{2h}^h$ maps coarse-grid values $\hat{V}_{i,t}^{2h}$ back to the fine grid:

$$\mathcal{I}_{2h}^h(\hat{V}_{i,t}^{2h}) = \begin{cases} \hat{V}_{i,t}^{2h}, & \text{if } t = 2nh, n \in \mathbb{N} \\ 0.5(\hat{V}_{i,t}^{2h} + \hat{V}_{i,t+h}^{2h}), & \text{if } t = (2n+1)h, n \in \mathbb{N} \end{cases}, \quad (9)$$

We define an optimality operator $B[\cdot]$ to evaluate the right-hand side of Eq. 4:

$$B[V_{i,t+h}^h] = \min_{u_i \in \mathcal{U}_i} \max \left\{ \max_{s \in [t,t+h]} c_i(\mathbf{x}_i(s)), V_i^h(\mathbf{x}_i(t+h), z_i(t+h), t+h) \right\}, \quad (10)$$

Using this operator, the fine-grid residuals are defined as $r_{i,t}^h = \hat{V}_{i,t}^h - B[\hat{V}_{i,t+h}^h]$. Our objective is to achieve $r_{i,t}^h \approx 0$ for all $(\mathbf{x}_i, z_i, t) \in \mathcal{X}_i \times \mathcal{Z}_i \times [0, T]$. To achieve this objective, we introduce the coarse-grid corrections $e_{i,t}^{2h}(\mathbf{x}_i, z_i)$ with grid size $2h$ and apply the two-level FAS method to solve the coarse-grid problem:

$$\mathcal{I}_h^{2h} r_{i,t}^h = B[\mathcal{I}_h^{2h}\hat{V}_{i,t+h}^h + e_{i,t+2h}^{2h}] - \left(\mathcal{I}_h^{2h}\hat{V}_{i,t}^h + e_{i,t}^{2h}\right) - \left(B[\mathcal{I}_h^{2h}\hat{V}_{i,t+h}^h] - \mathcal{I}_h^{2h}\hat{V}_{i,t}^h\right), \quad (11)$$

Simplifying Eq. 11, we obtain the update rule for the coarse-grid corrections:

$$e_{i,t}^{2h}(\mathbf{x}_i, z_i) = B[\mathcal{I}_h^{2h}\hat{V}_{i,t+h}^h + e_{i,t+2h}^{2h}] - B[\mathcal{I}_h^{2h}\hat{V}_{i,t+h}^h] - \mathcal{I}_h^{2h}r_{i,t}^h. \quad (12)$$

with terminal condition $e_{i,T}^{2h} = 0$. The full derivation of Eq. 11 is provided in Appendix C.3. The convergence proof and the complete description of Alg. 2 for multigrid refinement are provided in Appendix C.3. In practice, we train a neural network to approximate the correction term $e_{i,t}^{2h}(\mathbf{x}_i, z_i)$.

### 3.4 Epigraph-based Multigrid PINNs Learning

After introducing the key modules of our EMP framework, we now describe how they are integrated into a unified pipeline. In EMP, we leverage the structure of $V_i$ in Eq. 2 and introduce two neural networks $A_i : \mathcal{X} \times [0, T] \to \mathbb{R}$ and $B_i : \mathcal{X} \times [0, T] \to \mathbb{R}$ to approximate $V_i$ as

$$\hat{V}_i(\mathbf{x}_i, z_i, t) := \max \left\{ A_i(\mathbf{x}_i, t), B_i(\mathbf{x}_i, t) - z_i \right\}. \quad (13)$$

Here, $A_i$ predicts the worst-case future constraints violation, and $B_i$ approximates the equilibrium value $\vartheta_i$. If $A_i(\mathbf{x}_i, t) > 0$, then $\hat{V}_i > 0$ and $\vartheta_i(\mathbf{x}_i, t) = +\infty$, indicating safety constraints violation. Since EMP consists of two phases, which are epigraph-based PINN training and multigrid refinement, we adopt distinct sampling and training strategies for each.

**Epigraph-PINN sampling.** The first stage is data collection, where we adopt two sampling strategies. *PINN sampling* uniformly samples states and times $(\mathbf{x}_i, z_i, t)$ across the spatio-temporal domain to satisfy both the HJ equations ($\mathcal{N}$) and the boundary conditions ($\mathcal{B}$). Concretely, we construct two datasets $\mathcal{D}_\mathcal{N} := \{(\mathbf{x}_i, z_i, t)^{(n)} \in \mathcal{X} \times \mathcal{Z}_i \times [0, T]\}_{n=1}^{N_\mathcal{N}}$ and $\mathcal{D}_\mathcal{B} := \{(\mathbf{x}_i, z_i)^{(n)} \in \mathcal{X} \times \mathcal{Z}_i\}_{n=1}^{N_\mathcal{B}}$. *Rollout sampling*, in contrast, initializes trajectories from $(\mathbf{x}_i(0), z_i(0)) \in \mathcal{X}_{GT} \times \mathcal{Z}_i$ (where $\mathcal{X}_{GT}$

denotes the set of initial states obtained from the ground-truth solver) and propagates them forward in time using the current learned value $\hat{V}_i$. Along each trajectory, both values and value gradients are computed by discretizing Eq. 2 and Eq. 7 with a small time step $\Delta t$, producing the dataset $\mathcal{D}_R := \{(\mathbf{x}_i, z_i, t, V_i, \nabla_{\mathbf{x}_i} V_i)^{(n)}\}_{n=1}^{N_R}$, $N_R = T/\Delta t$. This procedure provides informative value–gradient pairs along closed-loop trajectories, which remain consistent with the system dynamics and serve as the key implicit supervision signal for training both value approximations $\hat{V}_i$ and their gradients $\nabla_{\mathbf{x}_i} \hat{V}_i$. Together, $\mathcal{D}_N, \mathcal{D}_B$, and $\mathcal{D}_R$ provide both global PDE satisfaction and local value-gradient signals, forming the foundation for epigraph-PINN training.

**Epigraph-PINN learning.** After data collection, the backbone of EMP is the training of an epigraph-based PINN, driven by a composite loss function. The first two terms are standard in PINN training: the *residual loss* enforces the HJ PDEs in Eq. 5, and the *boundary loss* in Eq. 6 enforces boundary conditions. However, as noted earlier, training with only these terms often leads to poor value approximations. To address this issue, we incorporate the value gradient dynamics module, which improves training with rollout-based supervision. This module contributes two additional loss terms: (i) aligning the learned value $\hat{V}_i$ with rollout-computed $V_i$, and (ii) aligning its gradient $\nabla_{\mathbf{x}_i} \hat{V}_i$ with rollout-computed $\nabla_{\mathbf{x}_i} V_i$. The full loss is

$$Loss(\hat{V}) := \sum_{i=1}^{2} \sum_{\mathcal{D}_N} \|\mathcal{N}(\hat{V}_i, \mathbf{x}_i, z_i, t)\|_1 + \sum_{\mathcal{D}_B} C_1 \|\mathcal{B}(\hat{V}_i, \mathbf{x}_i, z_i)\|_1$$
$$+ \sum_{\mathcal{D}_R} C_2 \|\hat{V}_i - V_i\|_1 + \sum_{\mathcal{D}_R} C_3 \|\nabla_{\mathbf{x}_i} \hat{V}_i - \nabla_{\mathbf{x}_i} V_i\|_1, \tag{14}$$

where $C_1, C_2, C_3 > 0$ balance the loss terms. To stabilize training, we adopt a curriculum scheme (Bansal & Tomlin, 2021): the network is first pretrained to satisfy boundary conditions, then progressively trained on states from an expanding time window starting at the terminal time, which helps capture the underlying physics of HJ PDEs.

**Multigrid refinement.** Once epigraph-PINN training is completed, we further improve the accuracy of value approximations through multigrid refinement. Using the rollout sampling strategy, we construct $\mathcal{D}_M := \{(\mathbf{x}_i, z_i, t)^{(n)}\}_{n=1}^{N_M}$ ($N_M = T/h$) with the current learned value $\hat{V}_i$. At each refinement stage, $\mathcal{D}_M$ is used to minimize the fine-grid residuals $r_{i,t}^h$ for all $t \in [0, T]$.

**Control policy extraction.** After training, EMP converts them into executable control policies. This involves a two-step procedure: First, we solve the epigraph-based optimization problem in Eq. 3 by minimizing $z_i$ subject to $V_i(\mathbf{x}_i, z_i, t) \leq 0$, which ensures $\hat{V}_i$ with respect to safety conditions. Second, given such $\hat{V}_i$, we compute safe control policies for Player $i$ by maximizing the Hamiltonian $\mathcal{H}_i$. The complete algorithm is provided in Alg. 1 of Appendix C.2.

## 4 EXPERIMENTS AND RESULTS DISCUSSION

In this section, we try to answer these three key questions: **1). How does our method perform under comparable computational budgets and training data compared to state-of-the-art (SOTA) baselines? 2). What impact do different activation functions have on the safety of control policies via learned values? 3). How do rollout sampling and multigrid refinement affect overall performance?** To answer these questions, we evaluate learned models on closed-loop trajectories and measure safety outcomes across three linear and nonlinear case studies.

**Benchmarks.** The first case models an uncontrolled intersection scenario with 5D systems (where the 5D domain includes the 4D physical state and time, consistent with the 9D and 13D cases), where two vehicles pass an intersection while avoiding collisions. The second case addresses narrow road avoidance with 9D systems, similar to prior zero-sum game formulations (Bansal & Tomlin, 2021), where both vehicles must adjust their speeds and headings to avoid collisions. The third case considers non-cooperative two-drone collision avoidance with 13D systems, where two drones fly along their respective 3D trajectories and optimize their controls to pass through a shared airspace without collisions. To the best of our knowledge, this represents the highest-dimensional HJ problem (Fridovich-Keil et al., 2020a). Fig. 9 in Appendix B illustrates the three scenarios, while Appendix B also provides detailed problem settings and model training procedures.

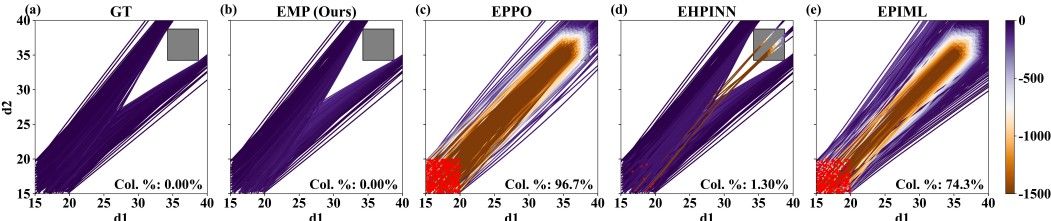

Figure 2: (a-e) The two-vehicle closed-loop trajectories (projected onto $d_1$–$d_2$) under $\mathcal{X}_{GT}$ calculated using Ground Truth, EMP, EPPO, EHPINN, and EPIML in Case 1. The color bar shows Player 1's equilibrium values. Gray boxes mark collision zones. Red dots indicate initial states with avoidable collisions.

**Baselines.** We evaluate EMP against four representative epigraph-based methods. Epigraph PPO (EPPO) (So & Fan, 2023) integrates the epigraph technique into PPO for optimal control. Its extensions to zero-sum (So et al., 2024) and cooperative games (Zhang et al., 2025) essentially reformulate multi-agent interactions as single-agent control. Epigraph Hybrid PINN (EHPINN) (Zhang et al., 2024) combines PINNs with PMP-driven supervised trajectories, yielding safe policies but suffering from the cost and sensitivity of solving BVPs. Epigraph PIML (EPIML) (Tayal et al., 2025) integrates the epigraph technique with physics-informed ML (Bansal & Tomlin, 2021). Although developed for optimal control, it is conceptually aligned with epigraph-based PINNs and can be adapted to general-sum games. EPIML derives control policies by maximizing the Hamiltonian, but suffers from singular arc issues when the Hamiltonian is linear with respect to the control input (Cristiani & Martinon, 2010). Therefore, we follow the same setting as EMP by adding a quadratic control term to the Hamiltonian, which ensures a fair comparison across methods and prevents singular arcs. Ground Truth (GT) is obtained by reformulating state constraints as collision penalties and solving BVPs via PMP (Kierzenka & Shampine, 2001), which provides reference trajectories for evaluation. For all Hamiltonian-based methods (EMP, EPIML, and EHPINN), policies are executed using $\nabla_{\mathbf{x}_i} \hat{V}_i$ derived from the learned value functions, while EPPO follows its policy network outputs. To ensure fairness, all baselines are trained with comparable dataset sizes and computational budgets (see Table 1, case study settings in Appendix B, and model settings in Appendix C.4).

**Evaluation Metrics.** All three case studies involve preventing collisions in two-player interactions, so we evaluate model performance using collision rate (Col.%) as the safety metric. The collision rate indicates deviations from theoretical safety, which is zero when using ground-truth solutions computed via BVP solvers. In practice, value approximation errors in neural networks can lead to collisions. We define $Col.\% = N_{pred}/N_{gt}$, where $N_{\text{pred}}$ is the number of trajectories with collisions when using neural networks as closed-loop controllers, and $N_{\text{gt}}$ is the number of collision-free trajectories obtained from BVP solutions. For each case, $N_{\text{gt}} = 600$ and neural networks generate the trajectories starting from the same initial state space $\mathcal{X}_{GT}$.

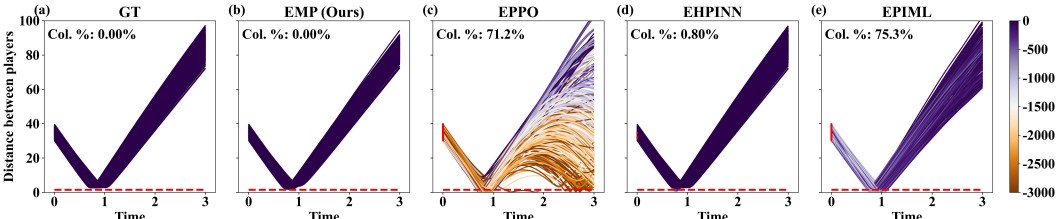

Figure 3: (a-e) The two-vehicle time-varying distance trajectories under $\mathcal{X}_{GT}$ calculated using Ground Truth, EMP, EPPO, EHPINN, and EPIML in Case 2. The red dashed line marks the minimum distance for collision.

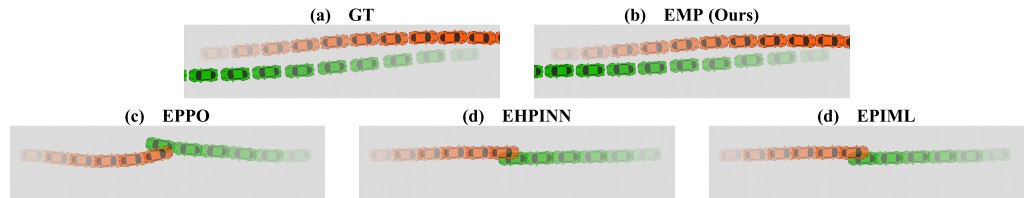

Figure 4: Case 2 visualization. (a): Ground truth safe trajectory. Trail opacity decays with time. (b-e): Trajectories generated by EMP, EPPO, EHPINN, and EPIML under the same initial conditions.

**Performance Evaluation.** We observe that EMP consistently achieves higher safety performance (lower collision rate) than all baselines across all case studies. The visualizations of Case 1-3 are reported in Fig. 2, 3 and 5, respectively. Fig. 4 and 6 illustrate a representative scenario from $\mathcal{X}_{GT}$ in

which all baseline methods fail to avoid a collision, while EMP succeeds. To explain this advantage, we analyze the limitations of the baselines. EPPO addresses multi-agent interactions (Zhang et al., 2025) using the centralized training with decentralized execution (CTDE) framework, where a centralized value function is trained with joint state–action information while decentralized policies act on local observations. Importantly, EPPO enforces a monotonic relationship between the centralized value and each agent's individual value through a max-aggregation scheme. This design is well-suited to cooperative settings, where agents optimize toward a shared objective (Sunehag et al., 2017; Rashid et al., 2020). However, in non-cooperative general-sum games, individual objectives differ, and the monotonicity assumption no longer holds. As a result, the centralized value fails to capture individual agent objectives, leading to unstable training and degraded safety performance in our tasks. EPIML adopts the PINN framework to approximate the value function and derives control policies by maximizing the resulting Hamiltonian. However, unlike EMP, EPIML does not incorporate the value gradient dynamics module or multigrid refinement. As a result, both the value function and its gradients are learned less accurately, which undermines the quality of the derived control policies and leads to degraded safety performance across all case studies. EHPINN incorporates value and gradient information from supervised data, leading to higher safety performance. However, it relies on BVP solvers following PMP to generate this supervision. This dependence introduces significant numerical challenges, e.g., sensitivity to initial trajectory guesses, singular arc issues, and increased computational cost. It is worth noting that EHPINN has lower safety performance and requires a higher training budget than our approach (see Table 1 in Appendix B). In contrast to EHPINN, EMP eliminates reliance on costly BVP solvers by combining self-supervised value-driven rollouts with multigrid refinement to approximate values. To further validate the performance of EMP, we compare the value contours generated by EMP against the ground truth in Appendix B.4.

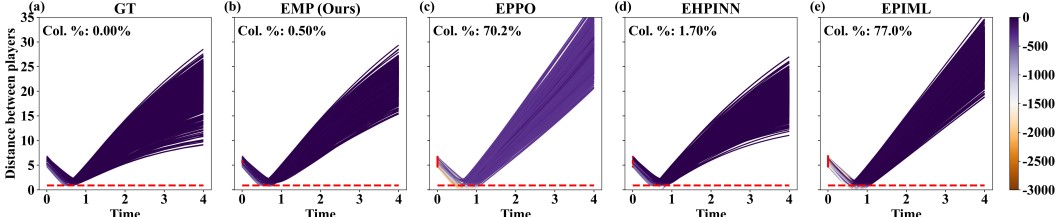

Figure 5: (a-e) The two-drone time-varying distance trajectories under $\mathcal{X}_{GT}$ calculated using Ground Truth, EMP, EPPO, EHPINN, and EPIML in Case 3.

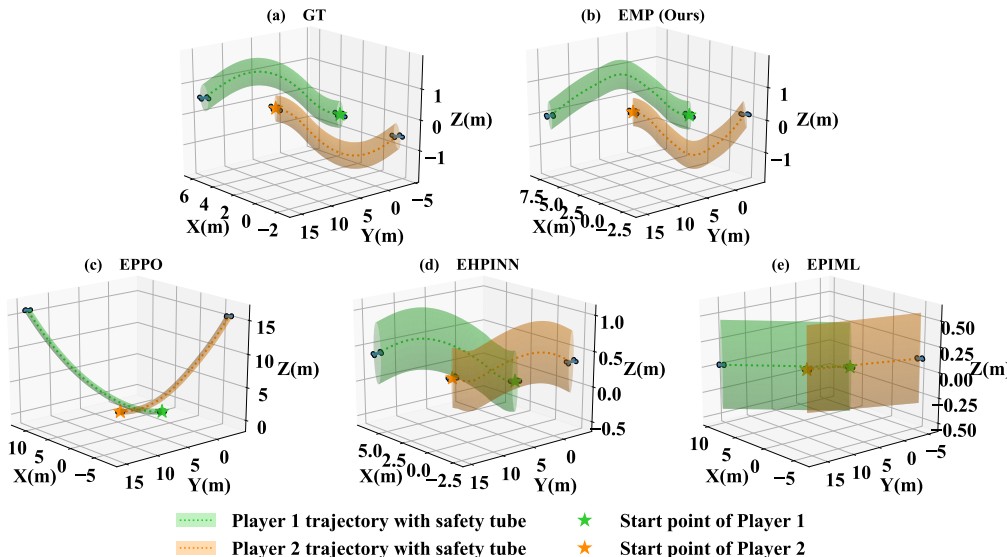

Figure 6: Case 3 visualization. (a) Ground-truth trajectory enclosed by the safety tube. (b-e) Trajectories generated by EMP, EPPO, EHPINN, and EPIML under the same initial conditions.

**Activation functions choice.** We evaluate how activation function choice affects EMP's performance, a factor previously emphasized in PINN-based learning methods (Bansal & Tomlin, 2021; Jagtap et al., 2020). Using the same EMP training setup across all case studies, we compare `tanh`, `relu`, and `sin` activations, reporting collision rates in Fig. 7. Across all tasks, `tanh` achieves the lowest

collision rate. Its smoothness and differentiability are particularly beneficial in PINN-based PDE solving, where automatic differentiation is used to compute value gradients and the PDE residuals (including boundary terms) is optimized via gradient descent. Smooth activations ensure stable gradient flow and reliable convergence, which is consistent with theoretical results showing that PINN training can converge on nonlinear PDEs with smooth solutions (Ito et al., 2021). While `sin` shares these smoothness properties, it is more sensitive to network parameter initialization (Sitzmann et al., 2020). In our experiments, this sensitivity leads `sin` to outperform `relu` in Case 2 and Case 3 but perform worse in Case 1 under the same initialization. In contrast, the piecewise-linear nature of `relu` suffers from zero-gradient regions, which can hinder PDE residuals minimization and even trigger gradient explosion in deeper networks.

**Ablation study.** We further evaluate the contribution of rollout sampling and multigrid refinement on the safety performance of EMP. For EMP without multigrid refinement, we train the PINN using Eq. 14 with `tanh` activation function, following the same procedure as EMP but omitting the multigrid stage in all case studies. For EMP without rollout sampling, we follow the same training procedure but uniformly sample $(\mathbf{x}_i, z_i, t)$ in the spatio-temporal domain and optimize only the PDE residuals and boundary condition losses, excluding the rollout-based value and gradient terms $\sum_{\mathcal{D}_R} |\hat{V}_i - V_i|_1$ and $\sum_{\mathcal{D}_R} |\nabla_{\mathbf{x}_i} \hat{V}_i - \nabla_{\mathbf{x}_i} V_i|_1$. The resulting value function is then refined with the same multigrid procedure as EMP. The detailed refinement settings are provided in Appendix B.

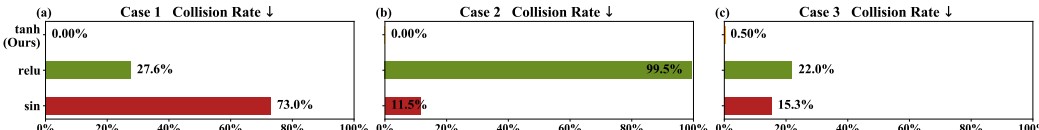

Figure 7: Summary of collision rates for EMP with `tanh`, `relu` and `sin` across all case studies.

The collision rates in Fig. 8 reveal two main insights. First, without multigrid refinement, EMP performs well in the 5D linear and 9D nonlinear systems but fails in the 13D highly nonlinear setting, where the collision rate rises to 75.3%. This indicates that rollout-driven value and gradient information substantially improve PINN training in 5D and 9D systems, but high-dimensional and nonlinear problems demand far more accurate value approximations. In the 13D case, deriving safe policies requires accurately capturing coupled dynamics across all 13 states, which is infeasible without high-precision value approximations. Multigrid refinement addresses this challenge by reducing fine-grid residuals through coarse-grid corrections, leading to more accurate value approximations and consistently higher safety performance. Second, removing rollout sampling causes EMP's epigraph-based PINN training to degrade to the performance level of EPIML, as it no longer benefits from self-supervised value–gradient signals along closed-loop trajectories. Although multigrid refinement can partially lower collision rates in this case, the resulting trajectories deviate from ground truth and incur higher training costs (see Table 2 in Appendix B, Fig. 15 and 16 in Appendix B.2). These findings highlight that rollout sampling is indispensable for generating accurate value–gradient pairs, while multigrid refinement is crucial for ensuring precision in complex dynamics. Together, both components are important for accurate value approximations and high safety performance in EMP.

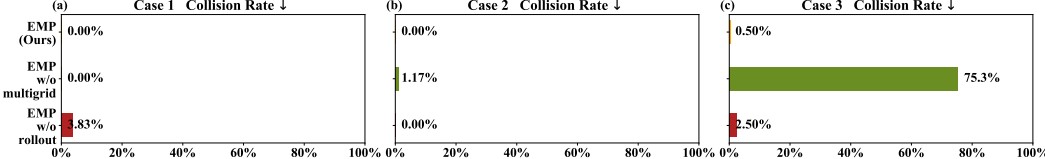

Figure 8: Summary of collision rates for EMP, EMP w/o multigird, and EMP w/o rollout in all case studies.

## 5 CONCLUSION

This work introduced EMP, a framework that integrates epigraph-based PINNs with multigrid refinement for solving two-player general-sum differential games with state constraints. Motivated by the need for accurate value and value gradient approximations without relying on supervised data, EMP leverages value gradient dynamics to design a value-driven rollout sampling strategy that provides self-supervised signals. Multigrid refinement further improves accuracy by propagating corrections across coarse and fine grids, reducing residual errors that PINNs alone cannot resolve. Across all case studies, EMP consistently achieves safer and more reliable policies than state-of-the-art baselines under comparable training budgets, demonstrating the effectiveness of combining

epigraph-based PINNs with multigrid refinement for safety-critical control. Future work will extend EMP to handle both environment-induced and multi-agent constraints, thereby enabling scalable and reliable deployment in general multi-agent systems.

## ETHICS STATEMENT

This work focuses on safe control policies in two-player general-sum differential games with state constraints. All experiments are conducted in simulation and do not involve human subjects or personal data.

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

## THE USE OF LARGE LANGUAGE MODELS

We used large language models to assist with polishing the writing and improving the overall readability of the paper.

## A  PROOFS AND DERIVATIONS

### A.1  EPIGRAPH-BASED VALUE GRADIENT DYNAMICS

In the problem statement, we have the following assumptions: $\mathcal{U}_i$ is compact and convex; $f_i$ and $c_i$ are Lipschitz continuous; $l_i$ and $g_i$ are Lipschitz continuous and bounded. We define the augmented state $y_i = (\mathbf{x}_i, z_i) \in \mathcal{X} \times \mathcal{Z}_i, \forall i = 1, 2$ for Player $i$ and augmented dynamics

$$\hat{\boldsymbol{f}}_i = \begin{bmatrix} \boldsymbol{f}_i(\mathbf{x}_i, u_i, u_{-i}) \\ -l_i(\mathbf{x}_i, u_i, u_{-i}) \end{bmatrix}.$$

Following Theorem 3.4 in Bokanowski et al. (2021) and Theorem 2 in Hermosilla & Zidani (2023), we introduce a value gradient $p_i = (p_i^x, p_i^z)$, where $p_i^x \in W^{1,1}([0,T]; \mathbb{R}^{d_x})$ and $p_i^z \in W^{1,1}([0,T]; \mathbb{R}^{d_z})$, a scalar $\lambda_i \in [0,1]$, a finite non-negative Borel measure $\mu_i$ on $[0,T]$, and a Borel measurable function $\gamma_i(t) = \nabla_{\mathbf{x}_i} c_i(\mathbf{x}_i)$. Define the augmented value gradient $q_i = (q_i^x, q_i^z)$ satisfying

$$q_i(t) := p_i(t) + \int_0^t \gamma_i(\tau)\,\mu_i(d\tau), \quad t \in [0,T), \qquad q_i(T) := p_i(T) + \int_0^T \gamma_i(\tau)\,\mu_i(d\tau). \quad (15)$$

Then the following conditions hold

(i) Nontriviality: $\lambda_i + \mu_i([0,T]) = 1$. (16)

(ii) Support of $\mu_i$: $\mathrm{supp}(\mu_i) \subseteq \{t \in [0,T] \mid c_i(\mathbf{x}_i) = 0\}$. (17)

(iii) Value gradient equation: $\dot{p}_i = \nabla_{y_i} q_i \cdot \hat{\boldsymbol{f}}_i$. (18)

(iv) Transversality Condition: $q_i(T) \in \lambda_i \nabla_{y_i}(g_i(x_i(T)) - z_i(T))$ (19)

(v) Hamiltonian Term: $q_i \cdot \hat{\boldsymbol{f}}_i = \max_{u_i \in \mathcal{U}_i} -q_i^x \cdot \boldsymbol{f}_i + q_i^z \cdot l_i$. (20)

*Remark* 1 (Interpretation). The measure $\mu_i$ acts as the Lagrange multiplier associated with the state constraints $c_i(\mathbf{x}_i)$. It is nonnegative and becomes positive at times when the constraints are active, e.g., $\mu_i(\{t\}) > 0$ when $c_i(\mathbf{x}_i(t)) = 0$ at time $t$. $\gamma$ denotes the gradient of the constraint function $c(\mathbf{x}_i)$ with respect to the state, $\gamma_i(t) = \nabla_{\mathbf{x}_i} c_i(\mathbf{x}_i)$. We note a special situation, following Remark 3.5 in Bokanowski et al. (2021): when the terminal cost $g_i$ holds and the state constraints are active, the Eq. 16 and 17 will be changed into $\lambda_i = 1$ and $\mu_i(\{t\}) > 0$ for all $t \in [0,T]$. Our problem settings satisfy this special situation.

Following Eq. 18 and 19, we can get

$$\dot{p}_i^x(t) = -q_i^x(t) \cdot \nabla_{\mathbf{x}_i} \boldsymbol{f}_i + q_i^z(t) \cdot \nabla_{\mathbf{x}_i} l_i, \quad (21)$$

$$\dot{p}_i^z(t) = 0 \quad \Rightarrow \quad p_i^z(t) = p_i^z(T), \quad (22)$$

$$q_i^x(T) = \lambda_i \nabla_{x_i} g_i(x_i(T)) \quad q_i^z(T) = -\lambda_i. \quad (23)$$

Since $\gamma_i(t) = \nabla_{\mathbf{x}_i} c_i(\mathbf{x}_i)$ and $q_i = (q_i^x, q_i^z)$ satisfies Eq. 15, we can have that

$$p_i^z(t) = q_i^z(t),\ p_i^z(T) = q_i^z(T),\ q_i^z(T) = -\lambda_i \quad \Rightarrow \quad q_i^z(t) = -\lambda_i. \quad (24)$$

Using $q_i(t) = p_i(t) + \int_0^t \gamma_i(\tau)\,\mu_i(d\tau),\ t \in [0,T)$ and Eq. 21, the dynamics for the augmented value gradient $q_i^x$ are

$$\dot{q}_i^x = -q_i^x \cdot \nabla_{\mathbf{x}_i} \boldsymbol{f}_i - \mu_i \nabla_{\mathbf{x}_i} c_i - \lambda_i \nabla_{\mathbf{x}_i} l_i. \quad (25)$$

At a time $t_k$ when the state constraints are active, i.e., $c_i(\mathbf{x}_i(t_k)) = 0$, the measure multiplier $\mu_i([t_k^-, t_k^+]) = 1$ and $\lambda_i = 0$, and $q_i^x$ has a jump characteristic

$$q_i^x(t_k^+) - q_i^x(t_k^-) = -\mu_i([t_k^-, t_k^+])\nabla_{\mathbf{x}_i} c_i\big(\mathbf{x}_i(t_k)\big) = -\nabla_{\mathbf{x}_i} c_i\big(\mathbf{x}_i(t_k)\big). \quad (26)$$

Otherwise, $\lambda_i = 1$ and $q_i^x$ evolves smoothly via Eq. 25 following $q_i^x(t_k^+) - q_i^x(t_k^-) = -\nabla_{\mathbf{x}_i} l_i$.

## B ADDITIONAL DETAILS IN THE EXPERIMENTS

To ensure a fair comparison between our method and all baselines, we use approximately the same number of sampled training data points. Detailed sampling strategies for each case study are provided in the following section, and the corresponding computational costs are summarized in Table 1 and 2. For improved training convergence, all neural network inputs are normalized to the range $[-1, 1]$.

Table 1: All learning methods computational costs and training dataset sizes across all case studies

| Case Study | Computational Cost (minutes) | EMP[1] | EPPO | EHPINN | EPIML |
|---|---|---|---|---|---|
| No. | Dataset Size | (phase 1 / phase 2) | (So & Fan, 2023) | (Zhang et al., 2024) | (Tayal et al., 2025) |
| | Supervised Data Acquisition | - | - | 93 | - |
| Case 1 | Model Training Cost | 225 / 20 | 257 | 168 | 288 |
| | Total Cost | **245** | 257 | 261 | 288 |
| | Training Dataset Size | 121k / 151k | 122k | 121k | 122k |
| | Supervised Data Acquisition | - | - | 135 | - |
| Case 2 | Model Training Cost | 277 / 8 | 300 | 204 | 347 |
| | Total Cost | **285** | 300 | 339 | 347 |
| | Training Dataset Size | 121k / 151k | 122k | 121k | 122k |
| | Supervised Data Acquisition | - | - | 310 | - |
| Case 3 | Model Training Cost | 379 / 161 | 552 | 290 | 600 |
| | Total Cost | **540** | 552 | 600 | 600 |
| | Training Dataset Size | 161k / 201k | 162k | 161k | 162k |

Table 2: Training costs (minutes) for EMP and ablation variants across all case studies

| Case Study No. | EMP | EMP w/o multigrid | EMP w/o rollout |
|---|---|---|---|
| Case 1 | 245 min | 225 min | 270 min |
| Case 2 | 285 min | 277 min | 322 min |
| Case 3 | 540 min | 540 min | 540 min |

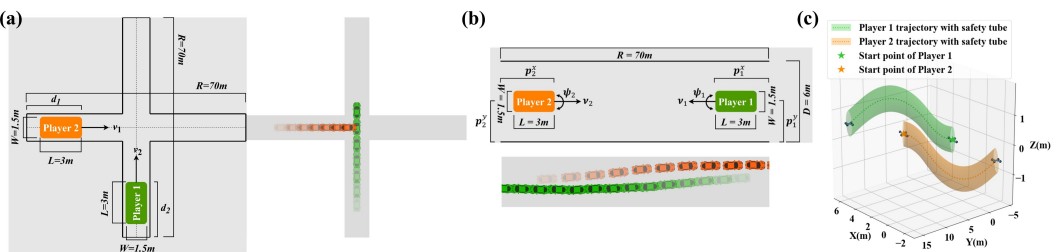

Figure 9: Case study schematics. (a) Case 1: uncontrolled intersection. (b) Case 2: narrow-road collision avoidance. (c) Case 3: two-drone collision avoidance.

### B.1 CASE 1: UNCONTROLLED INTERSECTION

In this game, each player's state is defined as $x_i := (d_i, v_i)$, where $d_i$ is the location along the road and $v_i$ is the speed. The shared dynamics follow

$$\dot{d}_i = v_i, \ \dot{v}_i = u_i,$$

where $u_i \in [-5, 10]$ m/s$^2$ represents acceleration/deceleration control. The instantaneous loss penalizes aggressive control inputs

$$l_i(\mathbf{x}_i, u_i) = u_i^2.$$

The state constraints enforce a safe distance between vehicles to avoid collisions

$$c_i(\mathbf{x}_i) = b\sigma(d_i)\sigma(d_{-i}) - \epsilon,$$

---

[1]Since EMP consists of two training phases: epigraph-based PINN and multigrid refinement, we report the computational cost and training dataset size for each phase in this table. For instance, in Case 1, the epigraph-based PINN requires 225 minutes of training with 121k data points, while the multigrid refinement needs 20 minutes of training with 151k data points.

where

$$\sigma(d) = \big(1 + \exp(-\gamma(d - R/2 + W/2))\big)^{-1}\big(1 + \exp(\gamma(d - (R + W)/2 - L))\big)^{-1},$$

with $\gamma = 1$ and $b = 100$ controlling the smoothness for the state constraint function. The parameters $R = 70$ m, $L = 3$ m, and $W = 1.5$ m denote road length, car length, and car width, respectively. The threshold $\epsilon = 24$ identifies if there exists violations of state constraints ($c_i(\mathbf{x}_i) > 0$). The terminal loss encourages both vehicles to cross the intersection while restoring nominal velocity

$$g_i(x_i) = -\alpha d_i(T) + (v_i(T) - \bar{v})^2,$$

with $\alpha = 10^{-6}$, $\bar{v} = 18$ m/s, and final time $T = 3$ s.

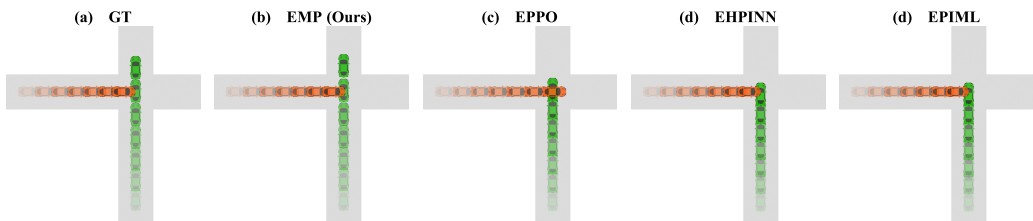

Figure 10: Case 1 visualization. (a) Ground truth safe trajectory. Trail opacity decays with time. (b-e) Trajectories generated by EMP, EPPO, EHPINN, and EPIML under the same initial conditions.

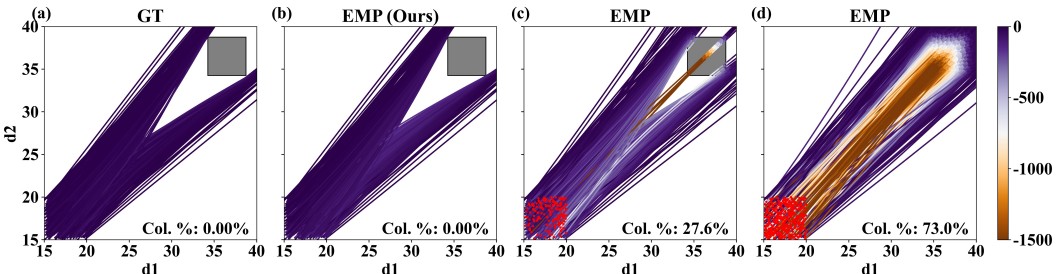

Figure 11: The two-vehicle closed-loop trajectories (projected onto $d_1$–$d_2$) under $\mathcal{X}_{GT}$ calculated using (a) Ground Truth and EMP with activation function (b) `tanh`, (c) `relu`, and (d) `sin` in Case 1.

**Data sampling and model training.** In this case, EPPO uses its policy network as a closed-loop controller, generating 1k trajectories per training epoch from initial states sampled in $\mathcal{X}_{GT} = [15, 20]$ m $\times [18, 25]$ m/s. Each trajectory consists of 122 steps at a time interval $\Delta t = 0.05$ s, yielding 122k data points in total, and training proceeds for 115 epochs with 10k gradient steps per epoch. EPIML uniformly samples 122k states from $\mathcal{X}_{HJ} = [15, 105]$ m $\times [15, 32]$ m/s and adopts the curriculum learning strategy from Bansal & Tomlin (2021). Training begins with 20k iterations to satisfy boundary conditions at the terminal time, followed by 330k gradient descent steps using states sampled from an expanding time window. EHPINN collects 500 ground truth trajectories (61k data points) uniformly sampled in $\mathcal{X}_{GT}$ by solving standard PMP and 60k states uniformly sampled in $\mathcal{X}_{HJ}$. Training includes a 20k pretraining stage for boundary conditions, followed by 200k iterations with both supervised and sampled states. EMP adopts a hybrid sampling strategy: (i) 60k states uniformly sampled from $\mathcal{X}_{HJ}$, and (ii) 61k states from 500 closed-loop trajectories initialized in $\mathcal{X}_{GT}$. EMP is trained with curriculum learning using the loss function in Eq. 14: 20k iterations for satisfaction of boundary conditions, followed by 300k gradient steps with state sampling under an expanding time window. The pretrained value network then generates 500 additional closed-loop trajectories from $\mathcal{X}_{GT}$, yielding 151k data points. A multigrid refinement stage follows with 10k gradient steps per epoch over 15 epochs. Both EMP without multigrid and EMP without rollout use the same training dataset size as EMP. For EMP without rollout, the model is refined using 10k gradient steps per epoch over 30 epochs. All methods uniformly sample $z_i \in [-1.05 \times 10^{-4}, 300]$.

Fig. 11 reports closed-loop trajectories obtained by EMP under different activation functions, highlighting the effect of activation function choice for model performance. Finally, Fig. 12 and 13 present trajectory predictions and visualizations from ablation studies, isolating the contributions of rollout sampling and multigrid refinement. Here, all models can avoid a collision, while the EMP without rollout fails.

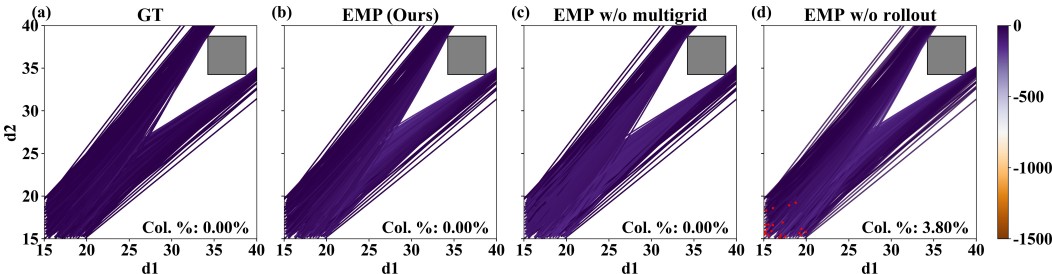

Figure 12: (a-d) The two-vehicle closed-loop trajectories (projected onto $d_1$–$d_2$) under $\mathcal{X}_{GT}$ calculated using Ground Truth, EMP, EMP without multigrid, and EMP without rollout in Case 1.

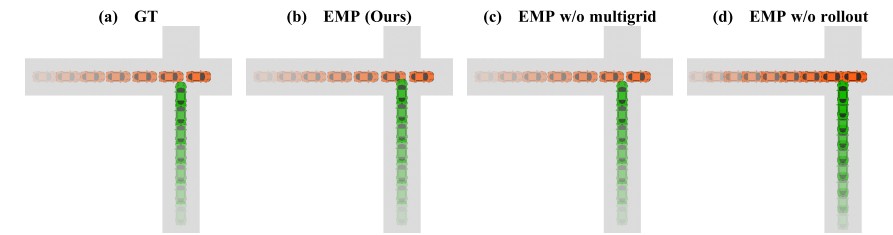

Figure 13: Case 1 visualization. (a) Ground truth safe trajectory. Trail opacity decays with time. (b-d) Trajectories generated by EMP, EMP without multigrid, and EMP without rollout under the same initial conditions.

### B.2 CASE 2: NARROW ROAD COLLISION AVOIDANCE

In this game, each player's state is $x_i := [p_i^x, p_i^y, \psi_i, v_i]^\top$, where $(p_i^x, p_i^y)$ is position, $\psi_i$ is orientation, and $v_i$ is speed. The system follows the unicycle dynamics

$$\dot{p}_i^x = v_i \cos(\psi_i), \; \dot{p}_i^y = v_i \sin(\psi_i), \; \dot{\psi}_i = \omega_i, \; \dot{v}_i = u_i,$$

with $\omega_i \in [-1, 1]$ rad/s and $u_i \in [-5, 10]$ m/s$^2$ representing angular velocity and acceleration controls, respectively. The instantaneous loss penalizes steering and acceleration control inputs

$$l_i(\mathbf{x}_i, \omega_i, u_i) = k\omega_i^2 + u_i^2,$$

with $k = 100$. The state constraints ensure collision avoidance

$$c_i(\mathbf{x}_i) = b\big(1 + \exp(-\gamma(\eta(\mathbf{x}_i) - S))\big)^{-1} - \epsilon,$$

where $\gamma = 1$, $b = 100$, $\epsilon = 50$. The distance between two vehicles is

$$S = \sqrt{((R - p_2^x) - p_1^x)^2 + (p_2^y - p_1^y)^2},$$

with road length $R = 70$ m and collision threshold $\eta = 1.5$ m. The terminal loss incentivizes vehicles to maintain lane position and nominal velocity

$$g_i(x_i) = -\alpha p_i^x(T) + (v_i(T) - \bar{v})^2 + (p_i^y(T) - \bar{p}^y)^2,$$

with $\alpha = 10^{-6}$, $\bar{v} = 18$ m/s, $\bar{p}^y = 4$ m, $T = 3$ s.

**Data sampling and model training.** The training setting is the same as Case 1 except for the data sampling domains. For EPPO, initial states are uniformly sampled from $\mathcal{X}_{GT} := [15, 20]$ m $\times$ $[2.25, 3.75]$ m $\times [-\pi/180, \pi/180]$ rad $\times [18, 25]$ m/s. For EPIML, 122k training states are drawn from $\mathcal{X}_{HJ} := [15, 90]$ m $\times [0, 6]$ m $\times [-0.15, 0.18]$ rad $\times [18, 25]$ m/s. For EHPINN, 500 ground truth trajectories (61k data points) are uniformly sampled in $\mathcal{X}_{GT}$ by solving standard PMP, and 60k states are uniformly sampled in $\mathcal{X}_{HJ}$. For EMP, we sample 60k states from $\mathcal{X}_{HJ}$ and 61k states from 500 closed-loop trajectories initialized in $\mathcal{X}_{GT}$. A multigrid refinement stage follows with 10k gradient steps per epoch over 5 epochs. Both EMP without multigrid and EMP without rollout use the same training dataset size as EMP. For EMP without rollout, the model is refined using 10k gradient steps per epoch over 30 epochs. All methods uniformly sample $z_i \in [-9.5 \times 10^{-5}, 300]$.

Fig. 14 illustrates closed-loop trajectories under different activation functions, highlighting how activation function choices affect model performance. Ablation results in Fig. 15 and 16 further

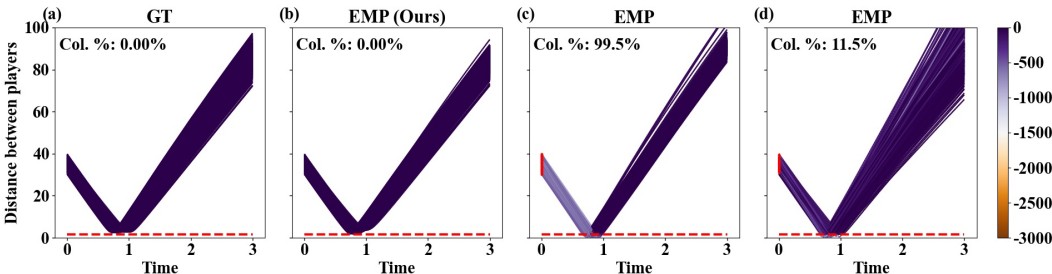

Figure 14: The two-vehicle time-varying distance trajectories under $\mathcal{X}_{GT}$ calculated using (a) Ground Truth and EMP with activation function (b) `tanh`, (c) `relu`, and (d) `sin` in Case 2.

isolate the contributions of rollout sampling and multigrid refinement. Without multigrid, EMP fails to prevent a collision; without rollout, EMP generates trajectories that are safe but deviate significantly from ground truth. Only the full EMP produces trajectories that are both collision-free and closely aligned with the ground-truth solution.

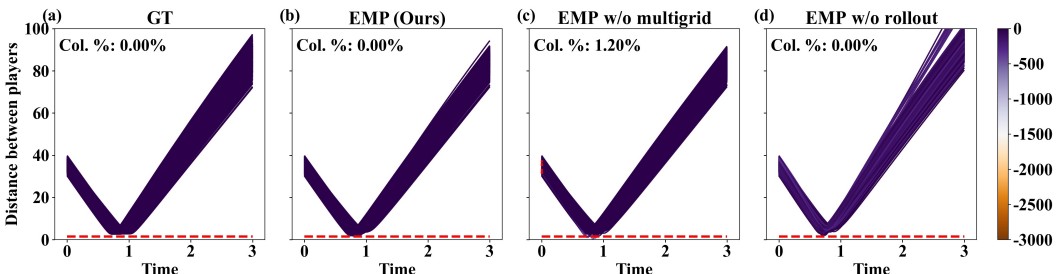

Figure 15: (a-d) The two-vehicle time-varying distance trajectories under $\mathcal{X}_{GT}$ calculated using Ground Truth, EMP, EMP without multigrid, and EMP without rollout in Case 2.

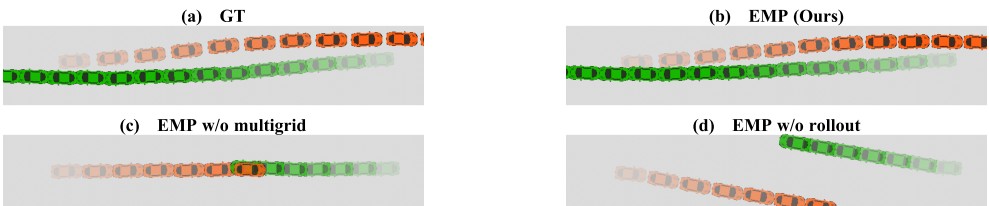

Figure 16: Case 2 visualization. (a) Ground truth safe trajectory. Trail opacity decays with time. (b-d) Trajectories generated by EMP, EMP without multigrid, and EMP without rollout under the same initial conditions.

### B.3 CASE 3: TWO-DRONE COLLISION AVOIDANCE

In this game, the system uses a 13D nonlinear model based on drone flight dynamics (Fridovich-Keil et al., 2020a), assuming zero yaw with respect to the global frame. Each player's state is $x_i := [p_i^x, p_i^y, p_i^z, v_i^x, v_i^y, v_i^z]^\top$, where $(p_i^x, p_i^y, p_i^z)$ is position and $(v_i^x, v_i^y, v_i^z)$ is velocity. The system dynamics are

$$\dot{p}_i^x = v_i^x, \ \dot{p}_i^y = v_i^y, \ \dot{p}_i^z = v_i^z, \ \dot{v}_i^x = g\tan(\psi_i), \ \dot{v}_i^y = -g\tan(\phi_i), \ \dot{v}_i^z = \tau_i - g,$$

where $u_i = (\psi_i, \phi_i, \tau_i)$ denotes roll, pitch, and thrust controls, with $\psi_i, \phi_i \in [-0.05, 0.05]$ rad, $\tau_i \in [7.81, 11.81]$ m/s$^2$, and $g = 9.81$ m/s$^2$. The instantaneous loss penalizes aggressive pitch, roll, and thrust control inputs

$$l_i(\mathbf{x}_i, \psi_i, \phi_i, \tau_i) = k_\psi \tan^2(\psi_i) + k_\phi \tan^2(\phi_i) + (\tau_i - g)^2,$$

with $k_\psi = k_\phi = 100$. The state constraints enforce a safe distance

$$c_i(\mathbf{x}_i) = b\big(1 + \exp(-\gamma(\eta(\mathbf{x}_i) - S))\big)^{-1} - \epsilon,$$

where $\gamma = 1$, $b = 100$, $\epsilon = 50$, and

$$S = \sqrt{((R_x - p_2^x) - p_1^x)^2 + ((R_y - p_2^y) - p_1^y)^2 + (p_2^z - p_1^z)^2},$$

with $R_x = 5$ m, $R_y = 5$ m, and collision threshold $\eta = 0.9$ m.

The terminal loss stabilizes position and velocity at the end of the horizon

$$g_i(x_i) = - \alpha p_i^x(T) - \alpha p_i^y(T) + (p_i^z(T) - \bar{p}_i^z)^2$$
$$+ (v_i^x(T) - \bar{v}_i^x)^2 + (v_i^y(T) - \bar{v}_i^y)^2 + (v_i^z(T) - \bar{v}_i^z)^2,$$

with $\alpha = 10^{-6}$, $\bar{p}_i^z = 0$ m, $\bar{v}_i^x = \bar{v}_i^y = \bar{v}_i^z = 0$ m/s, and $T = 4$ s.

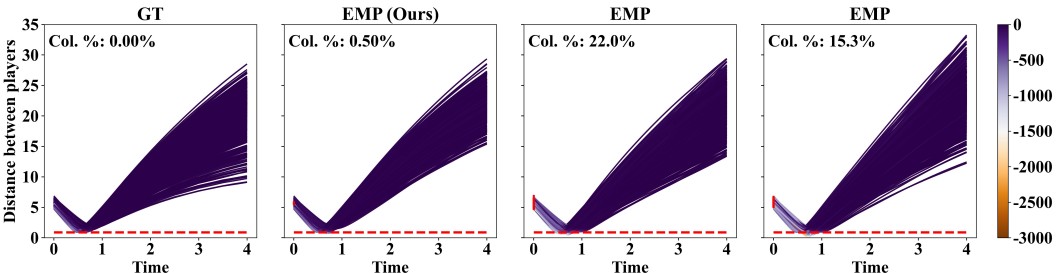

Figure 17: The two-drone time-varying distance trajectories under $\mathcal{X}_{GT}$ calculated using (a) Ground Truth and EMP with activation function (b) `tanh`, (c) `relu`, and (d) `sin` in Case 3.

**Data sampling.** For EPPO, initial states are uniformly sampled from $\mathcal{X}_{GT} := [0, 1]$ m $\times [0, 1]$ m $\times [-0.1, 0.1]$ m $\times [2, 4]$ m/s $\times [2, 4]$ m/s $\times [0, 0.1]$ m/s. Each trajectory has 162 data points (sampling interval 0.05 s), yielding 162k total data points. For EPIML, 162k states are sampled from $\mathcal{X}_{HJ} := [0, 15.5]$ m $\times [0, 15.5]$ m $\times [-1.8, 2]$ m $\times [0.3, 4.5]$ m/s $\times [0.3, 4.5]$ m/s $\times [-1.8, 1.8]$ m/s. For EHPINN, 500 PMP-driven trajectories (81k data points) are uniformly sampled in $\mathcal{X}_{GT}$, and 80k states are uniformly sampled in $\mathcal{X}_{HJ}$. For EMP, 80k states are drawn from $\mathcal{X}_{HJ}$ and 81k from 500 closed-loop trajectories initialized in $\mathcal{X}_{GT}$. The pretrained value network then generates 500 additional closed-loop trajectories from $\mathcal{X}_{GT}$, yielding 201k data points. Both EMP without multigrid and EMP without rollout use the same training dataset size as EMP. All methods uniformly sample $z_i \in [-3.1 \times 10^{-5}, 300]$.

**Model Training.** EPPO runs for 180 epochs with 10k gradient steps per epoch to refine both value and policy networks. EPIML begin with 20k iterations to satisfy boundary conditions, followed by 400k gradient descent steps with states sampled from an expanding time window. EMP and EMP without rollout apply multigrid refinement with 10k gradient steps per epoch over 80 epochs. EMP without multigrid is first trained for 20k iterations to satisfy boundary conditions, followed by 440k gradient updates with state sampling under an expanding time window. The remaining settings are the same as in Case 1.

Fig. 17 highlights how different activation functions influence the quality of closed-loop trajectories. Finally, ablation studies in Fig. 18 and 19 demonstrate the importance of rollout sampling and multigrid refinement: without either component, EMP deviates from ground truth or fails to remain collision-free, while the full method consistently produces safe and realistic trajectories.

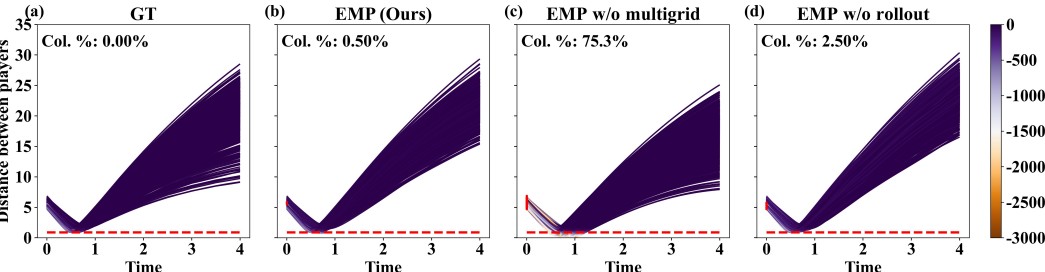

Figure 18: (a-d) The two-drone time-varying distance trajectories under $\mathcal{X}_{GT}$ calculated using Ground Truth, EMP, EMP without multigrid, and EMP without rollout in Case 3.

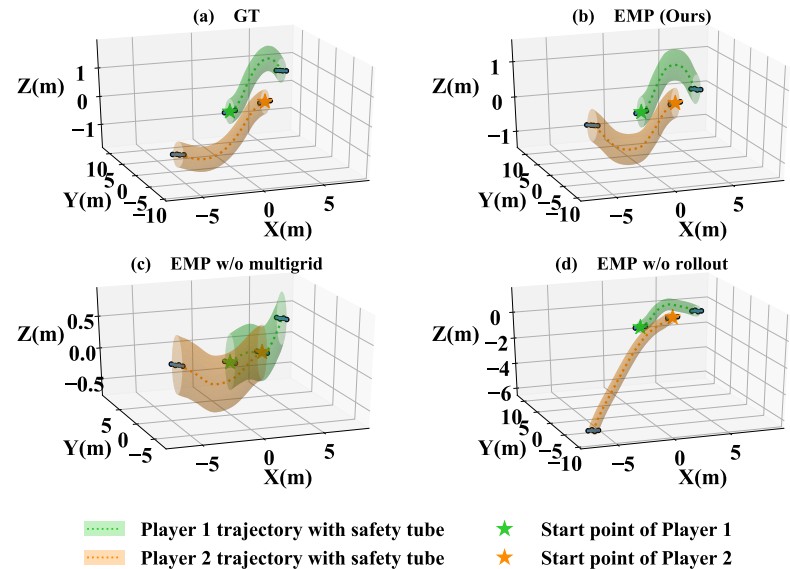

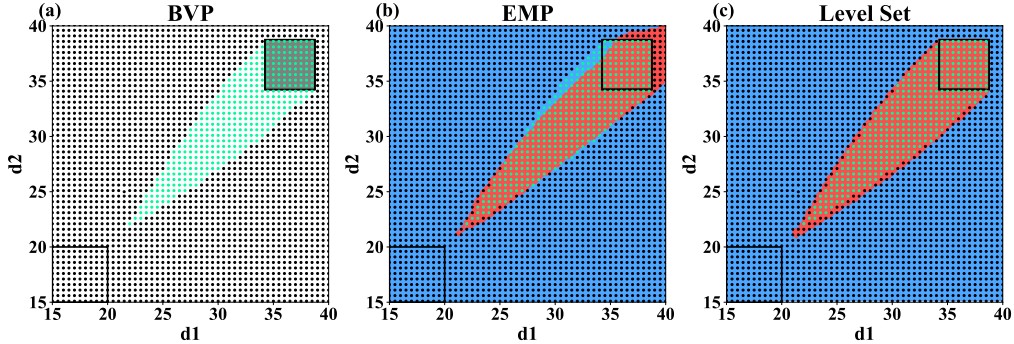

Figure 19: Case 3 visualization. (a) Ground-truth trajectory enclosed by the safety tube (b-d) Trajectories generated by EMP, EPPO, EHPINN, and EPIML under the same initial conditions.

Figure 20: (a) Ground-truth classification of initial states computed by BVP solvers, projected onto the $(d_1, d_2)$ frame. Black dots denote collision-free trajectories, and green dots indicate trajectories with collisions. (b-c) Value contours at $t_0$ obtained from EMP and the level set solver, respectively, are used to partition the state space into safe and unsafe zones. Blue regions correspond to safe states, while red regions correspond to unsafe states.

### B.4 BACKWARD REACHABLE SET COMPARISON

To further validate the performance of EMP, we revisit Case 1 and examine whether EMP consistently generates the backward reachable set (i.e., unsafe region) computed from the ground truth. From Eq. 2 and 3, the unsafe set is defined as $\{\mathbf{x} \in \mathcal{X}_{HJ} \mid V(\mathbf{x}, t_0) > 0\}$ at the initial time $t_0 = 0$ s. We fix the players' initial speeds at $v_{1,2} = 18$ m/s, sample initial positions $d_{1,2} \in [15, 40]$ m with resolution $\Delta d_{1,2} = 0.5$ m, and compute trajectories using BVP solvers following PMP to classify ground-truth safe and unsafe states. Figure 20a illustrates these results: green dots denote trajectories with collisions and black dots denote collision-free ones. Using the learned value $\hat{V}$, we compute the corresponding safe/unsafe sets and project the value contour onto the $(d_1, d_2)$ frame with $v_{1,2} = 18$ m/s and $t_0 = 0$. As shown in Fig. 20b, the unsafe zone predicted by EMP nearly coincides with the unsafe initial states identified by the BVP solvers.

It is well known that PMP provides necessary conditions for local optimality and yields open-loop policies (Bressan, 2010), whereas HJ equations guarantee global optimality and yield closed-loop policies. To mitigate this gap, we adopt multiple initial guesses when solving BVPs, which increases the chance of obtaining the global solutions, and we apply this treatment consistently across all case studies. In Case 1, for example, we give initial guess trajectories with four control pairs $\{(-5, -5), (-5, 10), (10, -5), (10, 10)\}$ m/s$^2$ for BVP solvers. These trajectories represent

different interactions for both players, where each player either yields or accelerates through the intersection, potentially leading to a different equilibrium. To address equilibrium multiplicity, we select the solution that is the best response of player values across these four interactions.

We empirically demonstrate that this treatment leads to consistent values between PMP and HJ solutions. Figure 20c visualizes the comparison in Case 1, where the value contours are projected onto $(d_1, d_2)$ with $v_{1,2} = 18\,\text{m/s}$ at $t_0$. To compute HJ-based values, we extend the level set solver of Bui et al. (2022) from zero-sum to general-sum games. Due to scalability limits (demonstrated up to 6D in Bui et al. (2022)), this computation is restricted to Case 1, where the state space is 5D. Using a spatial resolution of $dx = 0.3$, which is the highest resolution supported by our computing hardware, we compute values across $\mathcal{X}_{HJ}$. The resulting unsafe set (red) perfectly overlapped the unsafe states calculated by the BVP solvers, confirming the empirical consistency between PMP-based and HJ-based values. This conclusion is also supported by Zhang et al. (2024). Importantly, EMP produced nearly identical unsafe zones to the level set solver, further validating that EMP achieves accurate value approximations in safety-critical settings.

### B.5 ADDITIONAL ABLATION STUDY

In this section, we conduct additional ablation studies and comparisons to investigate the performance of EMP.

**Impact for fine-grid size $h$ choice.** Following Alg. 1, we adopt rollout sampling with a fixed time interval of $\Delta t = 0.05\,\text{s}$ to train the epigraph-based PINN model. For multigrid refinement, we vary the resolution of the fine grid using $h \in \{0.01\,\text{s}, 0.02\,\text{s}, 0.05\,\text{s}, 0.1\,\text{s}\}$. The overall training pipeline follows the settings described in Appendix B.1, B.2, and B.3. As shown in Fig. 21, the choice of $h$ has a significant effect on the safety performance of EMP. In Case 1, the collision rate is $0\%$ for small fine grid settings ($h = 0.01\,\text{s}$ and $h = 0.02\,\text{s}$), but increases when large fine grids are used. Case 2 exhibits the same pattern. In Case 3, where the dynamics are more nonlinear and the state space is higher dimensions, the influence of $h$ becomes more obvious: small fine grids ($h = 0.01\,\text{s}, 0.02\,\text{s}$) maintain high safety performance, while large fine grids ($h = 0.05\,\text{s}, 0.1\,\text{s}$) lead to significantly higher collision rates. The corresponding closed-loop trajectories are reported in Fig. 22, 23, and 24. In essence, the multigrid refinement leverages the dynamic programming principle to compute value functions in Eq. 4 along sampled rollouts, and smaller fine-grid resolutions yield more accurate value approximations and safer control policies. These results show that the grid size is a critical hyperparameter for the performance of EMP.

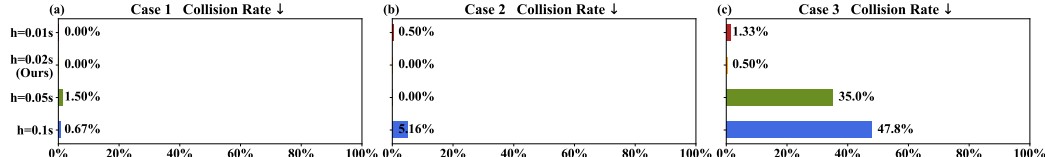

Figure 21: Summary of collision rates of EMP across all case studies with grid sizes $h \in \{0.01\,\text{s}, 0.02\,\text{s}, 0.05\,\text{s}, 0.1\,\text{s}\}$.

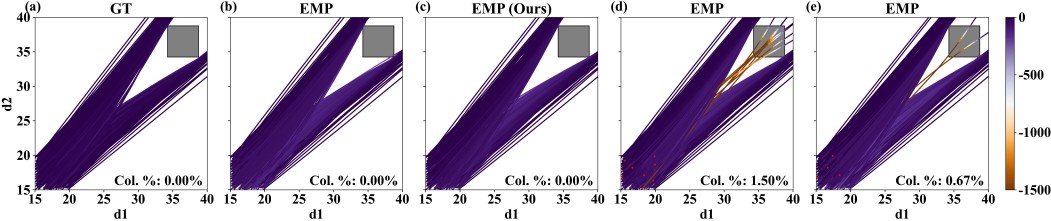

Figure 22: The two-vehicle closed-loop trajectories (projected onto $d_1$–$d_2$) under $\mathcal{X}_{GT}$, obtained using (a) Ground Truth, (b) EMP with grid sizes $h = 0.01\,\text{s}$, (c) EMP with grid sizes $h = 0.02\,\text{s}$, (d) EMP with grid sizes $h = 0.05\,\text{s}$, and (e) EMP with grid sizes $h = 0.1\,\text{s}$ in Case 1.

**Impact of weight parameters in loss function.** We balance the weights $C_1, C_2, C_3$ in Eq. 14 to ensure reliable safety performance of the epigraph-based PINN. To examine the effect of these weights, we evaluate three alternative settings: (1) large $C_1$ with balanced $C_2, C_3$; (2) large $C_2$ with balanced $C_1, C_3$; and (3) large $C_3$ with balanced $C_1, C_2$. The default *balanced* settings are reported

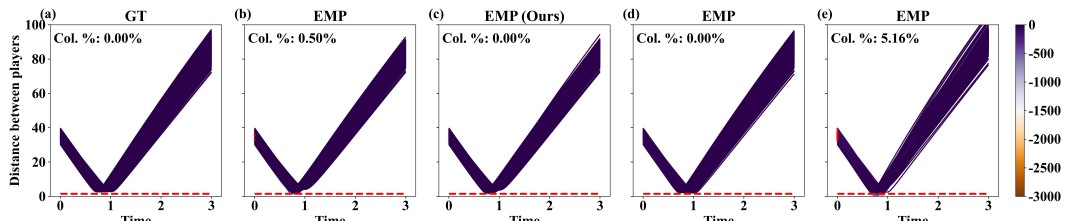

Figure 23: The two-vehicle time-varying distance trajectories under $\mathcal{X}_{GT}$, obtained using (a) Ground Truth, (b) EMP with grid sizes $h = 0.01\,\text{s}$, (c) EMP with grid sizes $h = 0.02\,\text{s}$, (d) EMP with grid sizes $h = 0.05\,\text{s}$, and (e) EMP with grid sizes $h = 0.1\,\text{s}$ in Case 2.

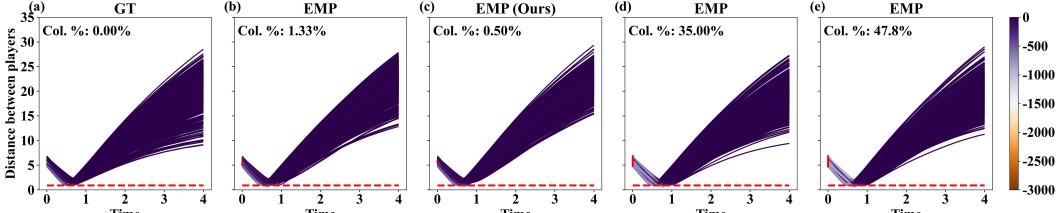

Figure 24: The two-drone time-varying distance trajectories under $\mathcal{X}_{GT}$, obtained using (a) Ground Truth, (b) EMP with grid sizes $h = 0.01\,\text{s}$, (c) EMP with grid sizes $h = 0.02\,\text{s}$, (d) EMP with grid sizes $h = 0.05\,\text{s}$, and (e) EMP with grid sizes $h = 0.1\,\text{s}$ in Case 3.

in Table 4 and the *large* weights are set to $\{4.5 \times 10^4, 6 \times 10^4, 2.5 \times 10^4\}$ for Case 1–3, respectively. The training procedure follows the configurations in Appendix B.1, B.2, and B.3. Using the learned models, we generate closed-loop trajectories and compute the corresponding collision rates. The results, summarized in Fig. 25, show that the balanced setting of $C_1, C_2, C_3$ yields the best safety performance across all case studies. The corresponding closed-loop trajectories are provided in Fig. 26, 27, and 28. Prior work has shown that imbalanced weights introduce stiffness into PINN training dynamics (Wang et al., 2021), causing the loss term with the largest weight to converge disproportionately faster than the others (Wang et al., 2022). This imbalance ultimately degrades safety performance when the learned models are used as closed-loop controllers in two-player interactions.

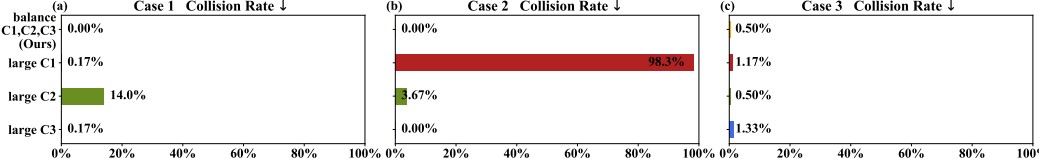

Figure 25: Summary of collision rates for EMP across all case studies with different weight $C_1, C_2, C_3$ settings.

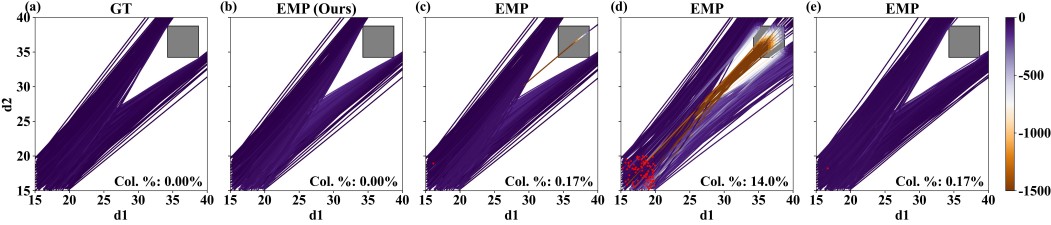

Figure 26: The two-vehicle closed-loop trajectories (projected onto $d_1$–$d_2$) under $\mathcal{X}_{GT}$, obtained using (a) Ground Truth, (b) EMP with balanced $C_1, C_2, C_3$, (c) EMP with large $C_1$ and balanced $C_2, C_3$, (d) EMP with large $C_2$ and balanced $C_1, C_3$, and (e) EMP with large $C_3$ and balanced $C_1, C_2$ in Case 1.

**Impact of dynamic noise.** In this paper, EMP relies on the deterministic system dynamics to generate rollouts for model training. To examine the robustness of the learned models, we introduce an additional Gaussian noise term into the time-invariant deterministic dynamics, expressed as $\dot{x}_i = f_i(x_i, u_i) + \text{noise}$. The system dynamics $f_i(x_i, u_i)$ follow the formulations in Appendix B.1, B.2, and B.3. We evaluate robustness under three noise levels, corresponding to zero-mean Gaussian disturbances with standard deviations of 0.1 (low), 0.5 (medium), and 1.0 (high). The training process is exactly the same as the EMP setup without dynamic noise. We report the resulting collision rates

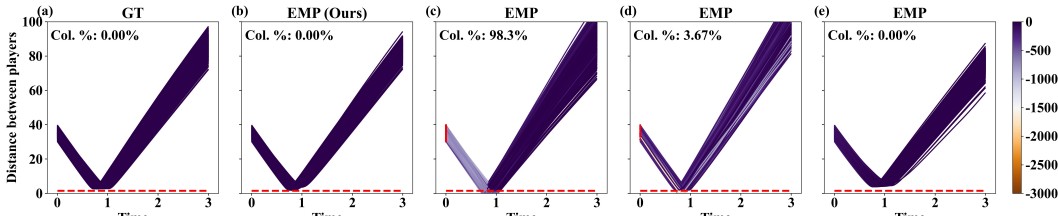

Figure 27: The two-vehicle time-varying distance trajectories under $\mathcal{X}_{GT}$, obtained using (a) Ground Truth, (b) EMP with balanced $C_1, C_2, C_3$, (c) EMP with large $C_1$ and balanced $C_2, C_3$, (d) EMP with large $C_2$ and balanced $C_1, C_3$, and (e) EMP with large $C_3$ and balanced $C_1, C_2$ in Case 2.

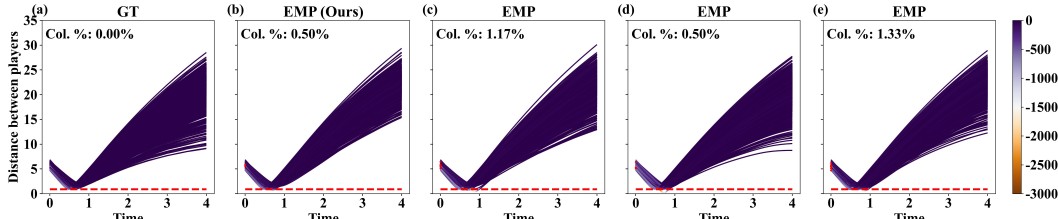

Figure 28: The two-drone time-varying distance trajectories under $\mathcal{X}_{GT}$, obtained using (a) Ground Truth, (b) EMP with balanced $C_1, C_2, C_3$, (c) EMP with large $C_1$ and balanced $C_2, C_3$, (d) EMP with large $C_2$ and balanced $C_1, C_3$, and (e) EMP with large $C_3$ and balanced $C_1, C_2$ in Case 3.

and closed-loop trajectories in Fig. 29, 30, 31, and 32. The results indicate that EMP maintains good safety performance under low dynamic noise in the 5D and 9D cases. In contrast, performance degrades in Case 3, which involves highly nonlinear dynamics and a higher-dimensional state space. In this setting, the added noise amplifies modeling errors in the learned value function, making the control policy more sensitive to rollout deviations. As expected, higher noise levels further increase collision rates across all cases. Although the epigraph-based value functions and the PDEs in this work are derived for deterministic systems and are not designed for stochastic dynamics, the experiments show that EMP retains good safety performance under small dynamic noise in both linear (5D) and nonlinear (9D) systems.

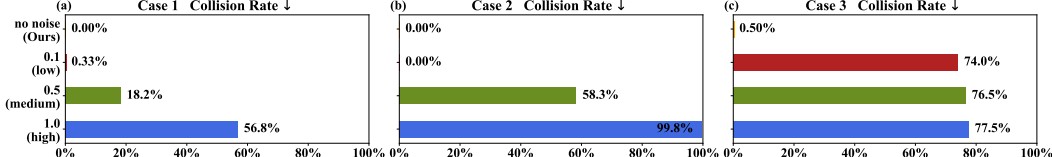

Figure 29: Collision rates for EMP across all case studies under Gaussian dynamic disturbances with standard deviations of 0 (no noise), 0.1 (low), 0.5 (medium), and 1.0 (high).

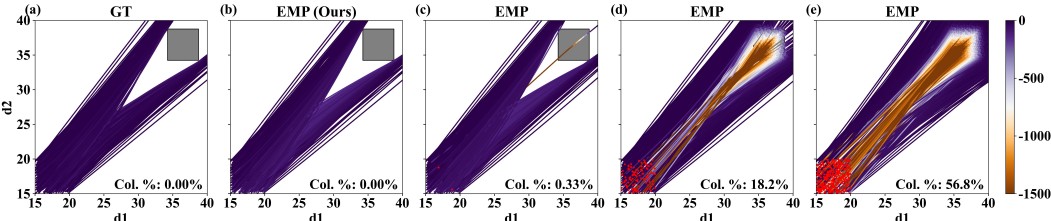

Figure 30: The two-vehicle closed-loop trajectories (projected onto $d_1-d_2$) under $\mathcal{X}_{GT}$, obtained using (a) Ground Truth, (b) EMP without dynamic noise, EMP with Gaussian dynamic noise of standard deviation (c) 0.1 (low), (d) 0.5 (medium), and (e) 1.0 (high) in Case 1.

**Comparison with general-sum multi-agent reinforcement learning (MARL).** In this paper, we focus on existing epigraph-based learning methods for comparison because the epigraph technique converts discontinuous value functions into continuous ones (Altarovici et al., 2013), thereby avoiding the difficulties associated with learning discontinuous values in two-player interactions (Zhang et al., 2024). In parallel, to compare EMP against a representative MARL baseline, we reformulate the state constraints as collision penalties and incorporate them into the instantaneous loss $l_i$ for $i = 1, 2$, and this reformulation is consistent with the one used in our ground-truth methods: the BVP and the

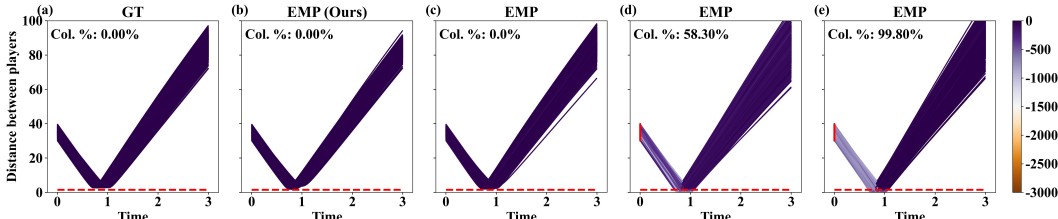

Figure 31: The two-vehicle time-varying distance trajectories under $\mathcal{X}_{GT}$, obtained using (a) Ground Truth, (b) EMP without dynamic noise, EMP with Gaussian dynamic noise of standard deviation (c) 0.1 (low), (d) 0.5 (medium), and (e) 1.0 (high) in Case 2.

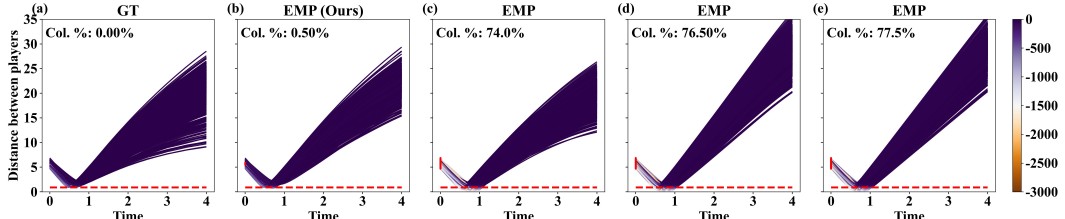

Figure 32: The two-drone time-varying distance trajectories under $\mathcal{X}_{GT}$, obtained using (a) Ground Truth, (b) EMP without dynamic noise, EMP with Gaussian dynamic noise of standard deviation (c) 0.1 (low), (d) 0.5 (medium), and (e) 1.0 (high) in Case 3.

level-set solver. For this comparison, we adopt the general-sum MARL algorithm MADDPG (Lowe et al., 2017). Because MADDPG optimizes cumulative rewards and does not enforce state constraints directly, the collision penalties supply the necessary immediate feedback near unsafe regions and encourage the emergence of collision-free behaviors. We design the following reward functions.

**Case 1:**

$$r_i = -(l_i(\mathbf{x}_i, u_i) + f_{\text{collision}})\,\Delta t,$$

where $f_{\text{collision}} = b\sigma(d_i)\sigma(d_{-i})$, $b = 10^4$, and $\Delta t = 0.05\,\text{s}$.

**Case 2:**

$$r_i = -(l_i(\mathbf{x}_i, \omega_i, u_i) + f_{\text{collision}})\,\Delta t,$$

where $f_{\text{collision}} = b\left(1 + \exp(-\gamma(\eta(\mathbf{x}_i) - S))\right)^{-1}$, $b = 10^4$, and $\Delta t = 0.05\,\text{s}$.

**Case 3:**

$$r_i = -(l_i(\mathbf{x}_i, \psi_i, \phi_i, \tau_i) + f_{\text{collision}})\,\Delta t,$$

with the same equation of $f_{\text{collision}}$ as in Case 2.

In all cases, the collision function satisfies $f_{\text{collision}} = 10^4$ when a collision occurs and $f_{\text{collision}} = 0$ otherwise. The remaining parameters follow the problem definitions in Appendix B.1, B.2, and B.3. The penalty magnitude is set to $b = 10^4$ to ensure collision avoidance.

Each player uses its own actor and critic network, and training is performed under the centralized-training–decentralized-execution (CTDE) paradigm. Detailed model settings appear in Appendix C.4. To ensure a fair comparison with EMP and the other baselines, MADDPG uses the same number of training data points. In the first training epoch, MADDPG generates 1k trajectories from initial states sampled in $\mathcal{X}_{GT}$. Each trajectory contains 122 steps with time interval $\Delta t = 0.05\,\text{s}$, yielding 122k data points, and performs 10k gradient updates. Beginning from the second epoch, MADDPG generates 500 new trajectories to refresh the replay buffer, then samples 1k trajectories (again with $\Delta t = 0.05\,\text{s}$) involving 122k data points for training and performs another 10k gradient updates per epoch until the training is completed.

We compare the safety performance of MADDPG with the BVP solutions and EMP across all case studies, with the results reported in Fig. 33, Fig. 34, and Fig. 35. The experimental results show that MADDPG fails to produce safe control policies in safety-critical two-player interactions. This outcome is consistent with the difficulty MADDPG faces in handling large collision penalties and emphasizes the importance of the epigraph formulation for achieving reliable safety performance in such settings.

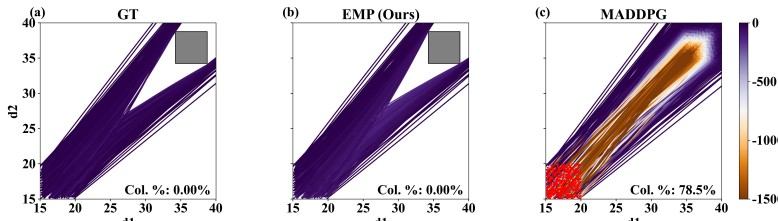

Figure 33: The two-vehicle closed-loop trajectories (projected onto $d_1$–$d_2$) under $\mathcal{X}_{GT}$, obtained using (a) Ground Truth, (b) EMP, and (c) MADDPG in Case 1.

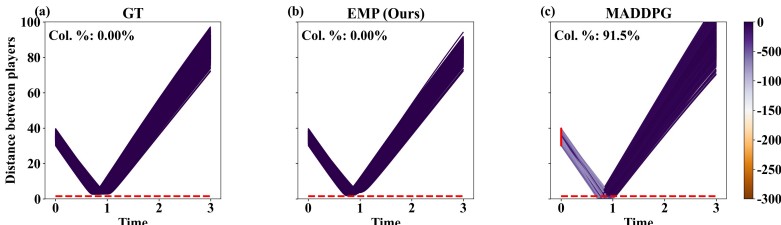

Figure 34: The two-vehicle time-varying distance trajectories under $\mathcal{X}_{GT}$, obtained using (a) Ground Truth, (b) EMP, and (c) MADDPG in Case 2.

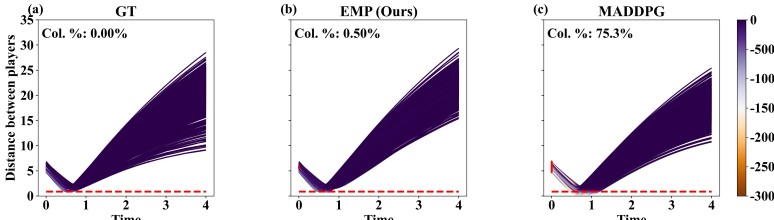

Figure 35: The two-drone time-varying distance trajectories under $\mathcal{X}_{GT}$, obtained using (a) Ground Truth, (b) EMP, and (c) MADDPG in Case 3.

**Comparison with iterative LQ solver.** For this comparison, we use the iterative LQ method (Fridovich-Keil et al., 2020a), an efficient numerical solver, to further verify the performance of EMP. We first reformulate the state constraints as collision penalties and incorporate them into the instantaneous loss, and this reformulation (rather than the epigraphic technique) is consistent with the settings used for MADDPG, the BVP solver, and the level-set method. To accelerate the computation of iterative LQ, we use the Julia implementation available at the public repository[2] and specify our problem within this framework.

In this experiment, we use a time interval of $\Delta t = 0.001\,\mathrm{s}$ for Case 1, 2 and 3. The instantaneous loss is defined as follows.

**Case 1:**
$$\tilde{l}_i = l_i(\mathbf{x}_i, u_i) + f_{\text{collision}},$$
where $f_{\text{collision}} = b\sigma(d_i)\sigma(d_{-i})$, $b = 10^4$.

**Case 2:**
$$\tilde{l}_i = l_i(\mathbf{x}_i, \omega_i, u_i) + f_{\text{collision}},$$
where $f_{\text{collision}} = b\left(1 + \exp(-\gamma(\eta(\mathbf{x}_i) - S))\right)^{-1}$, $b = 10^4$.

**Case 3:**
$$\tilde{l}_i = l_i(\mathbf{x}_i, \psi_i, \phi_i, \tau_i) + f_{\text{collision}},$$
with the same equation of $f_{\text{collision}}$ as in Case 2.

All other components of the problem (e.g., dynamics, terminal loss) follow the settings in Appendix B.1, B.2, and B.3. The BVP solver and level-set method use the same problem formulation, making the comparison consistent across all ground-truth techniques.

Given the same initial states sampled from $\mathcal{X}_{GT}$ for 600 trajectories in three cases, iterative LQ runs substantially faster than the BVP solver (e.g., ∼6 minutes compared to ∼100 minutes for the BVP

---

[2]https://github.com/JuliaGameTheoreticPlanning/iLQGames.jl

solver in Case 1). However, iterative LQ has lower convergence rates than the BVP solver in all cases: 81.0% vs. 98.8% in Case 1, 85.7% vs. 99.8% in Case 2, 40.2% vs. 99.9% in Case 3. Note: the 600 test trajectories solved by BVP are all converged solutions, while the reported BVP convergence rates are computed over 1k trajectories. The resulting closed-loop trajectories and collision statistics are shown in Fig. 36, 37 and 38. The results indicate that iterative LQ exhibits worse safety performance than both the BVP solver and EMP. This outcome is expected: the value function in Fridovich-Keil et al. (2020c) is continuous, while our formulation induces the nonsmooth value functions due to the large collision penalty. In addition, unlike BVP solvers, which support adaptive mesh refinement to improve accuracy near nonsmooth regions, iterative LQ uses a fixed time discretization determined before optimization and cannot refine it during the backward-forward passes. These limitations make iterative LQ less robust when handling nonsmooth value functions arising from large collision penalties.

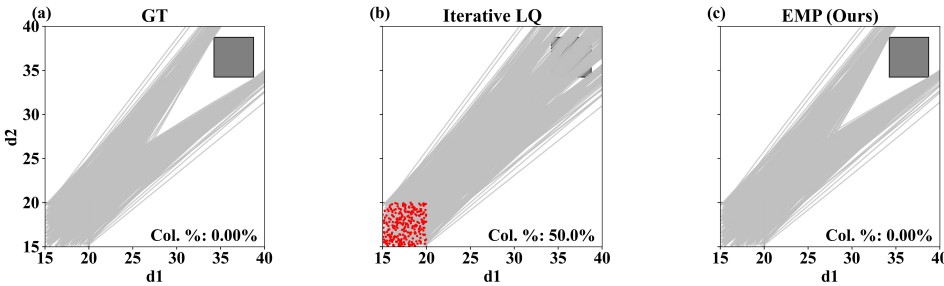

Figure 36: The two-vehicle closed-loop trajectories (projected onto $d_1$–$d_2$) under $\mathcal{X}_{GT}$, obtained using (a) Ground Truth, (b) iterative LQ, and (c) EMP in Case 1.

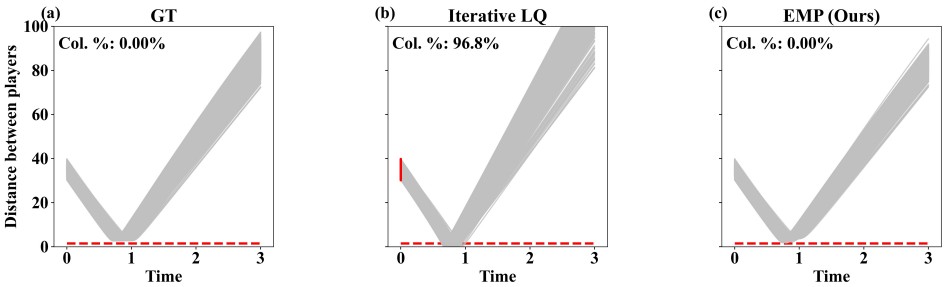

Figure 37: The two-vehicle time-varying distance trajectories under $\mathcal{X}_{GT}$, obtained using (a) Ground Truth, (b) iterative LQ, and (c) EMP in Case 2.

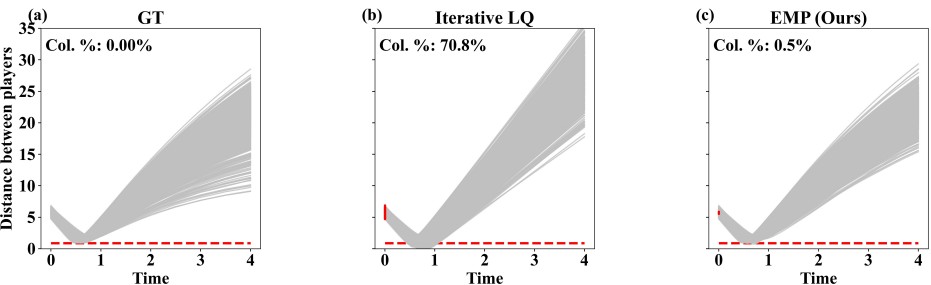

Figure 38: The two-drone time-varying distance trajectories under $\mathcal{X}_{GT}$, obtained using (a) Ground Truth, (b) iterative LQ, and (c) EMP in Case 3.

**Equilibrium verification of learned model.** To evaluate whether the learned model approximates the Nash equilibrium, we assess its value and action predictions on 600 test trajectories generated by the ground-truth solutions in all cases. For each sampled state-time pair $(x, t)$, we use the learned model to obtain the predicted value and action, and compare them against the corresponding ground-truth

Table 3: Prediction errors on ground-truth Nash equilibrium trajectories. The normalized MAE is computed using the admissible domain of each variable (e.g., control bounds or the value-function domain).

| Case | Metrics | MAE | rMAE | Range | normalized MAE |
|------|---------|-----|------|-------|----------------|
| Case 1 | $\mathbb{E}[\,\lvert \vartheta - \hat{\vartheta} \rvert\,]$ | 2.914 | 0.356 | $[-177.53, 0]$ | 0.016 |
| | $\mathbb{E}[\,\lvert u - \hat{u} \rvert\,]$ | 1.289 | 0.769 | $[-5, 10]$ | 0.086 |
| Case 2 | $\mathbb{E}[\,\lvert \vartheta - \hat{\vartheta} \rvert\,]$ | 2.364 | 0.652 | $[-22.77, -0.055]$ | 0.104 |
| | $\mathbb{E}[\,\lvert \omega - \hat{\omega} \rvert\,]$ | 0.047 | 0.584 | $[-1, 1]$ | 0.024 |
| | $\mathbb{E}[\,\lvert u - \hat{u} \rvert\,]$ | 0.550 | 0.642 | $[-5, 10]$ | 0.037 |
| Case 3 | $\mathbb{E}[\,\lvert \vartheta - \hat{\vartheta} \rvert\,]$ | 3.041 | 0.066 | $[-826.45, -0.19]$ | 0.004 |
| | $\mathbb{E}[\,\lvert \psi - \hat{\psi} \rvert\,]$ | 0.013 | 0.270 | $[-0.05, 0.05]$ | 0.130 |
| | $\mathbb{E}[\,\lvert \phi - \hat{\phi} \rvert\,]$ | 0.014 | 0.289 | $[-0.05, 0.05]$ | 0.140 |
| | $\mathbb{E}[\,\lvert \tau - \hat{\tau} \rvert\,]$ | 0.780 | 0.080 | $[7.81, 11.81]$ | 0.195 |

quantities. We report three metrics: the mean absolute error (MAE),

$$\mathbb{E}\big[\,\lvert F - \hat{F} \rvert\,\big],$$

the relative MAE,

$$\frac{\mathbb{E}\big[\,\lvert F - \hat{F} \rvert\,\big]}{\mathbb{E}\big[\,\lvert F \rvert\,\big]},$$

and the normalized MAE,

$$\frac{\mathbb{E}\big[\,\lvert F - \hat{F} \rvert\,\big]}{\text{range}},$$

where $F$ denotes the ground-truth value or action, $\hat{F}$ denotes the model prediction, and "range" refers to the admissible domain of each variable (e.g., control bounds or the value-function domain) computed by the ground-truth method. Since the ground-truth controller is itself a feedback Nash equilibrium, prediction accuracy on these equilibrium trajectories provides a direct and quantitative measure of how closely the learned model matches the equilibrium value and policy.

Table 3 summarizes the errors for the 5D, 9D, and 13D systems. In the 5D case, the value MAE is 2.914 (1.6% of the value range $[-177.53, 0]$), and the action MAE is 1.289 (8.6% of the control span $[-5, 10]$). In the 9D case, the action errors remain small (0.047 for $\omega$ and 0.550 for $u$, corresponding to 2.4% and 3.7% of their admissible domains), while the value MAE is 2.364 (10% of its value range $[-22.77, -0.055]$). For the 13D case, the value approximation is highly accurate (3.041, i.e., 0.4% of the range $[-826.45, -0.19]$), and the action errors for $\psi$, $\phi$, and $\tau$ fall within 13-20% of their feasible intervals.

Across all three case studies, the learned model reproduces the equilibrium value and action mappings with bounded error on the equilibrium trajectories. Combined with the closed-loop trajectories generated by the learned model (Fig. 4, Fig. 6, Fig. 10, Fig. 13, Fig. 16, and Fig. 19), these results indicate that the learned model can approximate the Nash equilibrium for values and actions.

### B.6 DISCUSSION ABOUT EPIGRAPH TECHNIQUE

To better understand the epigraph technique, we begin with the illustration in Fig. 39. Figure 39a shows a value function $\vartheta(x, t)$ that is finite only within a certain range of $x$ and becomes unbounded outside this region. To address this discontinuity, we introduce an auxiliary state variable $z$ that replaces $\vartheta(x, t)$ on the vertical axis and define an augmented value function $V(x, z, t)$, shown in Fig. 39b. The function $V(x, z, t)$ is Lipschitz continuous, and its sub-zero level set represents the epigraph of the original value function $\vartheta(x, t)$. This construction yields a continuous value function for the HJ equation, avoiding the numerical difficulties caused by discontinuities in the original formulation. We next present a toy example and provide the detailed computations to illustrate the epigraph-based HJ PDE, including the full verification procedure for checking whether a value function satisfies this PDE.

**Toy case.** We consider a general-sum differential game with two players moving in a one-dimensional setting. The players start from $x_1(0)$ and $x_2(0)$ at initial time $t_0$ and move toward each other with

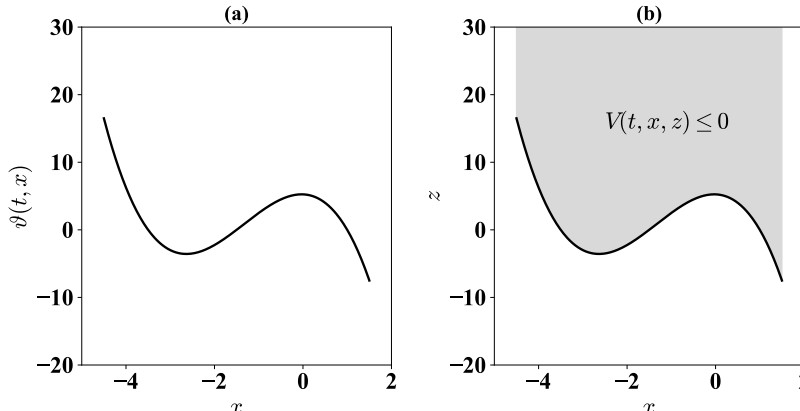

Figure 39: Given initial time t, (a) shows state-constrained value function $\vartheta(x, t)$ for each initial state x, and (b) shows the epigraph of $\vartheta(x, t)$ which is characterized by an augmented value function $V(x, z, t)$ (Lee, 2022).

speeds $u_1$ and $u_2$, respectively, where the admissible controls satisfy $u_1 \in [0, a]$ and $u_2 \in [-a, 0]$. The state constraint is defined as $c_1(\mathbf{x}) = c_2(\mathbf{x}) = x_1 - x_2 \leq 0$, which enforces that Player 1 must not cross to the right of Player 2. Here $\mathbf{x} = (x_1, x_2)$ denotes the joint state of the two players. The instantaneous loss is $l_1(\mathbf{x}) = l_2(\mathbf{x}) = 0$. The terminal loss is $g_1(x_1(T)) = -x_1(T)$ and $g_2(x_2(T)) = x_2(T)$, where each player seeks to maximize its final position while satisfying the state constraint. When the two players meet ($x_1 = x_2$), their speeds reduce to zero. A schematic illustration of this toy example is provided in Fig. 40.

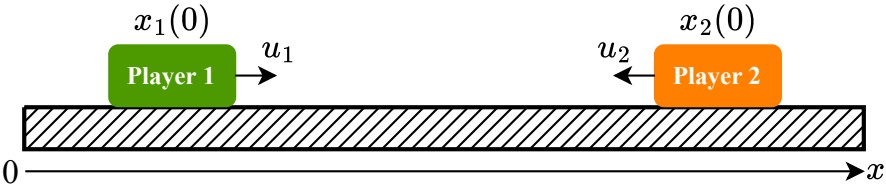

Figure 40: Schematic illustration of the toy case for general-sum games

**Computation results.** In this game, we know that the value of Player 1 satisfying

$$\vartheta_1(\mathbf{x}, t) = \min_{u_1 \in \mathcal{U}_1} g_1(x_1(T)) = \begin{cases} -\dfrac{x_1 + x_2}{2}, & if\ x_2 \geq x_1,\ \dfrac{x_2(0) - x_1(0)}{2a} \leq T - t_0 \\ -(x_1 + a(T - t)), & if\ x_2 \geq x_1,\ \dfrac{x_2(0) - x_1(0)}{2a} > T - t_0 \end{cases}$$

The value of Player 2 satisfies

$$\vartheta_2(\mathbf{x}, t) = \min_{u_2 \in \mathcal{U}_2} g_2(x_2(T)) = \begin{cases} \dfrac{x_1 + x_2}{2}, & if\ x_2 \geq x_1,\ \dfrac{x_2 - x_1}{2a} \leq T - t_0 \\ x_2 - a(T - t), & if\ x_2 \geq x_1,\ \dfrac{x_2 - x_1}{2a} > T - t_0 \end{cases}$$

where $\mathcal{U}_1$ and $\mathcal{U}_2$ are a set of measurable control policies for Player 1 and Player 2, respectively.

Therefore, we write out the epigraph-based HJ PDE for Player $i, \forall i = 1, 2$ as follows

$$\max \left\{ c_i(\mathbf{x}) - V_i(\mathbf{x}, z_i, t), \partial_t V_i - \mathcal{H}_i(\mathbf{x}, z_i, t, \nabla_{\mathbf{x}} V_i, \partial_{z_i} V_i) \right\} = 0,$$

where $\mathcal{H}_i$ is Hamiltonian and satisfies

$$\mathcal{H}_i = \max_{u_i \in \mathcal{U}_i} \left\{ -\partial_{\mathbf{x}} V_i \cdot \boldsymbol{f}_i - \partial_{z_i} V_i \cdot l_i \right\} = \max_{u_i \in \mathcal{U}_i} \left\{ -\partial_{\mathbf{x}} V_i \cdot \boldsymbol{f}_i \right\},$$

The boundary condition is denoted as $V_i(\mathbf{x}, z_i, T) = \max \left\{ c_i(\mathbf{x}), g_i(T) - z_i(T) \right\}$.

Next, we simplify the epigraph-based HJ PDE and have that

$$\max\left\{x_1(t) - x_2(t) - V_i(\mathbf{x}, z_i, t), \partial_t V_i - \mathcal{H}_i\right\} = 0, \tag{27}$$

For Player 1's Hamiltonian, we have that

$$\mathcal{H}_1 = \max_{u_1 \in \mathcal{U}_1}\left\{-\partial_{x_1} V_1 \cdot u_1 - \partial_{x_2} V_1 \cdot u_2^*\right\},$$

$$= \begin{cases} -\partial_{x_1} V_1 \cdot a - \partial_{x_2} V_1 \cdot u_2^*, & \partial_{x_1} V_1 < 0 \Longrightarrow u_1 = a \\ -\partial_{x_2} V_1 \cdot u_2^*, & \partial_{x_1} V_1 > 0 \Longrightarrow u_1 = 0 \end{cases}$$

Similarly, we have the Hamiltonian for Player 2

$$\mathcal{H}_2 = \max_{u_2 \in \mathcal{U}_2}\left\{-\partial_{x_1} V_2 \cdot u_1^* - \partial_{x_2} V_2 \cdot u_2\right\},$$

$$= \begin{cases} -\partial_{x_1} V_2 \cdot u_1^*, & \partial_{x_2} V_2 < 0 \Longrightarrow u_2 = 0 \\ -\partial_{x_1} V_2 \cdot u_1^* + \partial_{x_2} V_2 \cdot a & \partial_{x_2} V_2 > 0 \Longrightarrow u_2 = -a \end{cases}$$

We then write out the value function for Player 1, which is

$$V_1(\mathbf{x}, z_1, t) = \min_{u_1 \in \mathcal{U}_1} \max \left\{ \max_{s \in [t,T]} c_1(\mathbf{x}(s)), g_1(x_1(T)) - z_1 \right\},$$

$$= \max\left\{x_1(T) - x_2(T), -x_1(T) - z_1\right\}.$$

Specifically, $V_1(\mathbf{x}, z_1, t)$ should be equal to

$$\begin{cases} x_1(T) - x_2(T), \; if \; x_2 \geq x_1, \dfrac{x_2(0) - x_1(0)}{2a} \leq T - t_0, x_1(T) - x_2(T) \geq -x_1(T) - z_1 \; \text{(condition 1)} \\[2mm] -x_1(T) - z_1, \; if \; x_2 \geq x_1, \dfrac{x_2(0) - x_1(0)}{2a} \leq T - t_0, x_1(T) - x_2(T) < -x_1(T) - z_1 \; \text{(condition 2)} \\[2mm] x_1(T) - x_2(T), \; if \; x_2 \geq x_1, \dfrac{x_2(0) - x_1(0)}{2a} > T - t_0, x_1(T) - x_2(T) \geq -x_1(T) - z_1 \; \text{(condition 3)} \\[2mm] -x_1(T) - z_1, \; if \; x_2 \geq x_1, \dfrac{x_2(0) - x_1(0)}{2a} > T - t_0, x_1(T) - x_2(T) < -x_1(T) - z_1 \; \text{(condition 4)} \\[2mm] x_1(T) - x_2(T), \; if \; x_2 < x_1, x_1(T) - x_2(T) \geq -x_1(T) - z_1 \; \text{(condition 5)} \\[1mm] -x_1(T) - z_1, \; if \; x_2 < x_1, x_1(T) - x_2(T) < -x_1(T) - z_1 \; \text{(condition 6)} \end{cases}$$

$$= \begin{cases} 0, \; (\text{because } x_1(T) = x_2(T)), \; \text{if condition 1} \\[2mm] -\dfrac{x_1 + x_2}{2} - z_1, \; \text{if condition 2} \\[2mm] x_1 + u_1(T - t) - x_2 - u_2(T - t) = x_1 - x_2 - u_2^*(T - t), \; (u_1 = 0), \; \text{if condition 3} \\[1mm] -(x_1 + u_1(T - t)) - z = -(x_1 + a(T - t)) - z_1, \; (u_1 = a), \; \text{if condition 4} \\[1mm] x_1 + u_1(T - t) - x_2 - u_2(T - t) = x_1 - x_2 - u_2^*(T - t), \; (u_1 = 0), \; \text{if condition 5} \\[1mm] -(x_1 + u_1(T - t)) - z = -(x_1 + a(T - t)) - z_1, \; (u_1 = a), \; \text{if condition 6} \end{cases}$$

Similarly, the value function for Player 2 satisfies

$$V_2(\mathbf{x}, z_2, t) = \min_{u_2 \in \mathcal{U}_2} \max \left\{ \max_{s \in [t,T]} c_2(\mathbf{x}(s)), g_2(x_2(T)) - z_2 \right\},$$

$$= \max\left\{x_1(T) - x_2(T), x_2(T) - z_2\right\}.$$

Specifically, $V_2(\mathbf{x}, z_2, t)$ should be equal to

$$
\begin{cases}
x_1(T) - x_2(T), \text{ if } x_2 \geq x_1, \dfrac{x_2(0) - x_1(0)}{2a} \leq T - t_0, x_1(T) - x_2(T) \geq x_2(T) - z_2 \text{ (condition 1)} \\[2mm]
x_2(T) - z_2, \text{ if } x_2 \geq x_1, \dfrac{x_2(0) - x_1(0)}{2a} \leq T - t_0, x_1(T) - x_2(T) < x_2(T) - z_2 \text{ (condition 2)} \\[2mm]
x_1(T) - x_2(T), \text{ if } x_2 \geq x_1, \dfrac{x_2(0) - x_1(0)}{2a} > T - t_0, x_1(T) - x_2(T) \geq x_2(T) - z_2 \text{ (condition 3)} \\[2mm]
x_2(T) - z_2, \text{ if } x_2 \geq x_1, \dfrac{x_2(0) - x_1(0)}{2a} > T - t_0, x_1(T) - x_2(T) < x_2(T) - z_2 \text{ (condition 4)} \\[2mm]
x_1(T) - x_2(T), \text{ if } x_2 < x_1, x_1(T) - x_2(T) \geq x_2(T) - z_2 \text{ (condition 5)} \\[2mm]
x_2(T) - z_2, \text{ if } x_2 < x_1, x_1(T) - x_2(T) < x_2(T) - z_2 \text{ (condition 6)}
\end{cases}
$$

$$
=
\begin{cases}
0, \text{ (because } x_1(T) = x_2(T)), \text{ if condition 1} \\[2mm]
\dfrac{x_1 + x_2}{2} - z_2, \text{ if condition 2} \\[2mm]
x_1 + u_1(T - t) - x_2 - u_2(T - t) = x_1 - x_2 + u_1^*(T - t), \ (u_2 = 0), \text{ if condition 3} \\[2mm]
(x_2 + u_2(T - t)) - z_2 = (x_2 - a(T - t)) - z, \ (u_2 = -a), \text{ if condition 4} \\[2mm]
x_1 + u_1(T - t) - x_2 - u_2(T - t) = x_1 - x_2 + u_1^*(T - t), \ (u_2 = 0), \text{ if condition 5} \\[2mm]
(x_2 + u_2(T - t)) - z_2 = (x_2 - a(T - t)) - z_2, \ (u_2 = -a), \text{ if condition 6}
\end{cases}
$$

To verify whether $V_i(\mathbf{x}, z_i, t)$ is the solution to HJ PDE, we plug $V_i(t, \mathbf{x}, z_i)$ into Eq. 27. We first examine the boundary condition at $t = T$ and start with Player 1.

$$
V_1(\mathbf{x}, z_1, T) =
\begin{cases}
0 = x_1(T) - x_2(T), \text{ if condition 1} \\[2mm]
-\dfrac{x_1(T) + x_2(T)}{2} - z_1 = -x_1(T) - z_1, \text{ if condition 2} \\[2mm]
x_1(T) - x_2(T) - u_2^*(T - T) = x_1(T) - x_2(T), \text{ if condition 3} \\[2mm]
-(x_1(T) + a(T - T)) - z_1 = -x_1(T) - z_1, \text{ if condition 4} \\[2mm]
x_1(T) - x_2(T) - u_2^*(T - T) = x_1(T) - x_2(T), \text{ if condition 5} \\[2mm]
-(x_1(T) + a(T - T)) - z_1 = -x_1(T) - z_1, \text{ if condition 6}
\end{cases}
$$

Next, we examine the condition for Player 2

$$
V_2(\mathbf{x}, z_2, T) =
\begin{cases}
0 = x_1(T) - x_2(T), \text{ if condition 1} \\[2mm]
\dfrac{x_1(T) + x_2(T)}{2} - z_2 = x_2(T) - z_2, \text{ if condition 2} \\[2mm]
x_1(T) - x_2(T) + u_1^*(T - T) = x_1(T) - x_2(T), \text{ if condition 3} \\[2mm]
(x_2(T) + a(T - T)) - z_2 = x_2(T) - z_2, \text{ if condition 4} \\[2mm]
x_1(T) - x_2(T) + u_1^*(T - T) = x_1(T) - x_2(T), \text{ if condition 5} \\[2mm]
(x_2(T) + a(T - T)) - z_2 = x_2(T) - z_2, \text{ if condition 6}
\end{cases}
$$

The results show that $V_i(\mathbf{x}, z_i, t), \forall i = 1, 2$ satisfies the boundary condition.

We now examine the HJ PDE and start with Player 1.

*Condition 1:* $V_1(\mathbf{x}, z_1, t) = 0$, we can compute that $\partial_t V_1 = 0$, $\partial_{x_1} V_1 = 0$ and $\partial_{x_2} V_1 = 0$. Then it derives that $\mathcal{H}_1 = 0$. Since there is no violation of state constraint, $x_1(t) - x_2(t) \leq 0$. Hence, we have that

$$
\max\{x_1(t) - x_2(t) - 0, 0\} = 0,
$$

which shows that $V_1(\mathbf{x}, z_1, t)$ satisfies the HJ PDE under condition 1.

*Condition 2:* $V_1(\mathbf{x}, z_1, t) = -\frac{x_1(t) + x_2(t)}{2} - z_1$, we can compute that $\partial_t V_1 = 0$, $\partial_{x_1} V_1 = -\frac{1}{2}$ and $\partial_{x_2} V_1 = -\frac{1}{2}$. Then it derives that $u_1 = a$, $u_2 = -a$ and $\mathcal{H}_1 = -(-\frac{1}{2}) \cdot a - (-\frac{1}{2}) \cdot (-a) = 0$. (Note: Player 2 needs to force Player 1 to stop). Since there is no violation of state constraint,

$x_1(t) - x_2(t) \leq 0$. Also, $V_1(\mathbf{x}, z_1, t) = -\frac{x_1(t)+x_2(t)}{2} - z_1 \geq 0 \implies \frac{x_1(t)+x_2(t)}{2} + z_1 \leq 0$. Hence, we have that

$$\max\left\{x_1(t) - x_2(t) + \frac{x_1(t) + x_2(t)}{2} + z_1, 0\right\} = 0,$$

which shows that $V_1(\mathbf{x}, z_1, t)$ satisfies the HJ PDE under condition 2.

*Condition 3:* $V_1(\mathbf{x}, z_1, t) = x_1(t) - x_2(t) - u_2^*(T - t)$, we can compute that $\partial_t V_1 = u_2^*$, $\partial_{x_1} V_1 = 1$ and $\partial_{x_2} V_1 = -1$. Then it derives that $u_1 = 0$, $u_2 = u_2^*$ and $\mathcal{H}_1 = -(-1) \cdot u_2^* = u_2^*$. Hence, we have that

$$\max\left\{x_1(t) - x_2(t) - x_1(t) + x_2(t) + u_2^*(T - t), 0\right\} = 0,$$

which shows that $V_1(\mathbf{x}, z_1, t)$ satisfies the HJ PDE under condition 3. Note:

$$a_2^*(T - t) = \begin{cases} 0, \ if \ u_2^* = 0 \\ -a(T - t) < 0, \ if \ u_2^* = -a \end{cases}$$

*Condition 4:* $V_1(\mathbf{x}, z_1, t) = -(x_1(t) - u_2^*(T-t)) - z_1$, we can compute that $\partial_t V_1 = a$, $\partial_{x_1} V_1 = -1$ and $\partial_{x_2} V_1 = 0$. Then it derives that $u_1 = a$ and $\mathcal{H}_1 = -(-1) \cdot a = a$. Hence, we have that

$$\max\left\{x_1(t) - x_2(t) + (x_1(t) + a(T - t)) + z_1, 0\right\} = 0,$$

which shows that $V_1(\mathbf{x}, z_1, t)$ satisfies the HJ PDE under condition 4. Note:

$$-(x_1(t) + a(T - t)) - z_1 = -x_1(T) - z_1 > x_1(T) - x_2(T) > x_1(t) - x_2(t)$$

*Condition 5:* $V_1(\mathbf{x}, z_1, t) = x_1(t) - x_2(t) - u_2^*(T - t)$, we can compute that $\partial_t V_1 = u_2^*$, $\partial_{x_1} V = 1$ and $\partial_{x_2} V_1 = -1$. Then it derives that $a_1 = 0$, $u_2 = u_2^*$ and $\mathcal{H}_1 = -(-1) \cdot u_2^* = u_2^*$. Hence, we have that

$$\max\left\{x_1(t) - x_2(t) - x_1(t) + x_2(t) + u_2^*(T - t), 0\right\} = 0,$$

which shows that $V_1(\mathbf{x}, z_1, t)$ satisfies the HJ PDE under condition 5. Note:

$$a_2^*(T - t) = \begin{cases} 0, \ if \ u_2^* = 0 \\ -a(T - t) < 0, \ if \ u_2^* = -a \end{cases}$$

*Condition 6:* $V_1(\mathbf{x}, z_1, t) = -(x_1(t) + a(T-t)) - z_1$, we can compute that $\partial_t V_1 = a$, $\partial_{x_1} V_1 = -1$ and $\partial_{x_2} V_1 = 0$. Then it derives that $u_1 = a$ and $\mathcal{H}_1 = -(-1) \cdot a = a$. Hence, we have that

$$\max\left\{x_1(t) - x_2(t) + (x_1(t) + a(T - t)) + z_1, 0\right\} = 0,$$

which shows that $V_1(\mathbf{x}, z_1, t)$ satisfies the HJ PDE under condition 6. Note:

$$-(x_1(t) + a(T - t)) - z_1 = -x_1(T) - z_1 > x_1(T) - x_2(T) > x_1(t) - x_2(t)$$

Finally, we examine the condition for Player 2.

*Condition 1:* $V_2(\mathbf{x}, z_2, t) = 0$, we can compute that $\partial_t V_2 = 0$, $\partial_{x_1} V_2 = 0$ and $\partial_{x_2} V_2 = 0$. Then it derives that $\mathcal{H}_2 = 0$. Since there is no violation of state constraint, $x_1(t) - x_2(t) \leq 0$. Hence, we can have that

$$\max\left\{x_1(t) - x_2(t) - 0, 0\right\} = 0,$$

which shows that $V_2(\mathbf{x}, z_2, t)$ satisfies the HJ PDE under condition 1.

*Condition 2:* $V_2(\mathbf{x}, z_2, t) = \frac{x_1(t)+x_2(t)}{2} - z_2$, we can compute that $\partial_t V_2 = 0$, $\partial_{x_1} V_2 = \frac{1}{2}$ and $\partial_{x_2} V_2 = \frac{1}{2}$. Then it derives that $u_1 = a$, $u_2 = -a$ and $\mathcal{H}_2 = -\frac{1}{2} \cdot a - \frac{1}{2} \cdot (-a) = 0$. (Note: Player 1 needs to force Player 2 to stop). Since there is no violation of state constraint, $x_1(t) - x_2(t) \leq 0$. Also, $V_2(\mathbf{x}, z_2, t) = \frac{x_1(t)+x_2(t)}{2} - z_2 \geq 0 \implies -\frac{x_1(t)+x_2(t)}{2} + z_2 \leq 0$. Hence, we can have that

$$\max\left\{x_1(t) - x_2(t) - \frac{x_1(t) + x_2(t)}{2} + z_2, 0\right\} = 0,$$

which shows that $V_2(\mathbf{x}, z_2, t)$ satisfies the HJ PDE under condition 2.

*Condition 3:* $V_2(\mathbf{x}, z_2, t) = x_1(t) - x_2(t) + u_1^*(T - t)$, we can compute that $\partial_t V_2 = -u_1^*$, $\partial_{x_1} V_2 = 1$ and $\partial_{x_2} V_2 = -1$. Then it derives that $u_1 = u_1^*$, $u_2 = 0$ and $\mathcal{H}_2 = -1 \cdot u_1^* = -u_1^*$. Hence, we have that
$$\max\left\{x_1(t) - x_2(t) - x_1(t) + x_2(t) - u_1^*(T - t), 0\right\} = 0,$$
which shows that $V_2(\mathbf{x}, z_2, t)$ satisfies the HJ PDE under condition 3. Note:
$$-u_1^*(T - t) = \begin{cases} 0, & if \ u_1^* = 0 \\ -a(T - t) < 0, & if \ u_1^* = a \end{cases}$$

*Condition 4:* $V_2(\mathbf{x}, z_2, t) = (x_2(t) - a(T - t)) - z_2$, we can compute that $\partial_t V_2 = a$, $\partial_{x_1} V_2 = 0$ and $\partial_{x_2} V_2 = 1$. Then it derives that $u_2 = -a$ and $\mathcal{H}_2 = -1 \cdot (-a) = a$. Hence, we have that
$$\max\left\{x_1(t) - x_2(t) - (x_2(t) - a(T - t)) + z_2, 0\right\} = 0,$$
which shows that $V_2(\mathbf{x}, z_2, t)$ satisfies the HJ PDE under condition 4. Note:
$$(x_2(t) - a(T - t)) - z_2 = x_2(T) - z_2 > x_1(T) - x_2(T) > x_1(t) - x_2(t)$$

*Condition 5:* $V_2(\mathbf{x}, z_2, t) = x_1(t) - x_2(t) + u_1^*(T - t)$, we can compute that $\partial_t V_2 = -u_1^*$, $\partial_{x_1} V_2 = 1$ and $\partial_{x_2} V_2 = -1$. Then it derives that $u_1 = u_1^*$, $u_2 = 0$ and $\mathcal{H}_2 = -1 \cdot u_1^* = -u_1^*$. Hence, we have that
$$\max\left\{x_1(t) - x_2(t) - x_1(t) + x_2(t) - u_1^*(T - t), 0\right\} = 0,$$
which shows that $V_2(\mathbf{x}, z_2, t)$ satisfies the HJ PDE under condition 5. Note:
$$-u_1^*(T - t) = \begin{cases} 0, & if \ u_1^* = 0 \\ -a(T - t) < 0, & if \ u_1^* = a \end{cases}$$

*Condition 6:* $V_2(\mathbf{x}, z_2, t) = (x_2(t) - a(T - t)) - z_2$, we can compute that $\partial_t V_2 = a$, $\partial_{x_1} V_2 = 0$ and $\partial_{x_2} V_2 = 1$. Then it derives that $u_2 = -a$ and $\mathcal{H}_2 = -1 \cdot (-a) = a$. Hence, we have that
$$\max\left\{x_1(t) - x_2(t) - (x_2(t) - a(T - t)) + z_2, 0\right\} = 0,$$
which shows that $V_2(\mathbf{x}, z_2, t)$ satisfies the HJ PDE under condition 6. Note:
$$(x_2(t) - a(T - t)) - z_2 = x_2(T) - z_2 > x_1(T) - x_2(T) > x_1(t) - x_2(t)$$

In summary, this toy example demonstrates the full verification procedure for checking whether a value function satisfies the epigraph-based HJ PDE, thereby offering an intuitive understanding of its structure and behavior.

## C  IMPLEMENTATION DETAILS OF THE ALGORITHMS

### C.1  HARDWARE

We collect supervised equilibrium data by solving BVPs using the standard PMP. For implementation, we employ the standard `solve_bvp` solver from `scipy`[3] with an error tolerance of $5 \times 10^{-3}$ and a maximum of 1500 mesh nodes for all case studies. Numerical experiments of equilibrium data collection are conducted on a system with an Intel(R) Xeon E5-1620 v4 CPU (3.50 GHz) and an NVIDIA GTX TITAN X GPU (12 GB memory). Training for our method and all baselines is performed on an NVIDIA A100 GPU (40 GB memory). This setup ensures both the computational efficiency and the numerical precision required for large-scale simulations in deep learning–based control policy derivation for general-sum differential games.

---

[3] https://docs.scipy.org/doc/scipy/reference/generated/scipy.integrate.solve_bvp.html

## C.2 EMP ALGORITHM

We summarize the training procedure of Epigraph-based Multigrid PINNs (EMP) as follows

---

**Algorithm 1:** Epigraph-based Multigrid PINN

---

**Input:** number of initial states $N$, time horizon $\mathcal{T} = [0, T]$, learning rate $\alpha_l$, network parameter
$\quad\quad \theta$, $pretrain\_iterations$, $train\_iterations$

**Initialize:** epigraph-based PINN $\hat{V}_i$, epigraph-based PINN dataset $D_{\mathcal{N}} = \emptyset$, $D_{\mathcal{B}} = \emptyset$, rollout
$\quad\quad$ sampling dataset $D_R = \emptyset$, multigrid refinement dataset $D_M = \emptyset$

1   $\mathcal{D}_{\mathcal{B}} \leftarrow$ sample $(\mathbf{x}_i, z_i) \in \mathcal{X} \times \mathcal{Z}_i$, pretrain $\hat{V}_i$ at $pretrain\_iters$ to satisfy boundary condition in
    Eq. 6

2   set $iter = 0$, $num\_epoch = 0$

3   **while** $iter \leq train\_iterations$ **do**

4      $\mathcal{D}_{\mathcal{B}} \leftarrow$ sample $(\mathbf{x}_i, z_i) \in \mathcal{X} \times \mathcal{Z}_i$, $\mathcal{D}_{\mathcal{N}} \leftarrow$ sample $(\mathbf{x}, z_i) \in \mathcal{X} \times \mathcal{Z}_i$, $\forall i = 1, 2$

5      **for** $t \in \mathcal{T}$ **do**

6        $\mathcal{D}_{\mathcal{N}} \leftarrow$ sample $t \in [0, iter/train\_iterations * T]$

7      **end**

8      **if** *num_epoch mod 10 == 0* **then**

9        sampling $N$ initial states $(\mathbf{x}_i(0), z_i(0)) \in \mathcal{X}_{GT} \times \mathcal{Z}_i$

10       generate rollouts using learned value $\hat{V}_i$ to collect training dataset $\mathcal{D}_R$

11       $\mathcal{D}_R \leftarrow$ compute $\hat{V}_i$, $\nabla_{\mathbf{x}_i}\hat{V}_i$ numerically using Eq. 2 and Eq. 7

12      $num\_epoch \leftarrow num\_epoch + 1$

13      $\theta \leftarrow \theta - \alpha_l \nabla_\theta Loss(\hat{V})$

14      $iter \leftarrow iter + 1$

15 **end**

16 refine learned values $\hat{V}_i$ following Alg. 2

---

## C.3 MULTIGRID ALGORITHM

To solve a nonlinear problem $Au = f$, we let $A$ denote a nonlinear operator, $u$ the exact solution, $e = u - v$ the error, and $r = f - Av$ the residual. Building on this formulation, we rewrite Eq. 4 using an optimality operator $B[\cdot]$. Specifically, we define $Au$ as $A(V_{i,t}^h) = B[V_{i,t+h}^h] - V_{i,t}^h$ and have that

$$B[V_{i,t+h}^h] - V_{i,t}^h = 0 \;=\; f_{i,t}^h.$$

Following the first step of the FAS method, we define the fine-grid residual, let $Av$ be $A(\hat{V}_{i,t}^h) = B[\hat{V}_{i,t+h}^h] - \hat{V}_{i,t}^h$ and have that

$$r_{i,t}^h = f_{i,t}^h - \left(B[\hat{V}_{i,t+h}^h] - \hat{V}_{i,t}^h\right) = \hat{V}_{i,t}^h - B[\hat{V}_{i,t+h}^h].$$

We follow the second and third steps of FAS to compute each term

$$A^{2h}(\hat{V}_i^{2h} + e_i^{2h}) = B\left[\mathcal{I}_h^{2h}\hat{V}_{i,t+h}^h + e_{i,t+2h}^{2h}\right] - \left(\mathcal{I}_h^{2h}\hat{V}_{i,t}^h + e_{i,t}^{2h}\right),$$

$$A^{2h}(\hat{V}_i^{2h}) = B[\mathcal{I}_h^{2h}\hat{V}_{i,t+h}^h] - \mathcal{I}_h^{2h}\hat{V}_{i,t}^h, r_i^{2h} = \mathcal{I}_h^{2h}r_{i,t}^h . r_i^{2h} \quad\quad = \mathcal{I}_h^{2h}r_{i,t}^h.$$

We then solve the coarse-grid correction problem

$$A^{2h}(\hat{V}_i^{2h} + e_i^{2h}) = A^{2h}(\hat{V}_i^{2h}) + r_i^{2h},$$

$$\Rightarrow \quad B\left[\mathcal{I}_h^{2h}\hat{V}_{i,t+h}^h + e_{i,t+2h}^{2h}\right] - \left(\mathcal{I}_h^{2h}\hat{V}_{i,t}^h + e_{i,t}^{2h}\right) = B[\mathcal{I}_h^{2h}\hat{V}_{i,t+h}^h] - \mathcal{I}_h^{2h}\hat{V}_{i,t}^h + \mathcal{I}_h^{2h}r_{i,t}^h,$$

$$\Rightarrow \quad e_{i,t}^{2h}(\mathbf{x}_i, z_i) = B[\mathcal{I}_h^{2h}\hat{V}_{i,t+h}^h + e_{i,t+2h}^{2h}] - B[\mathcal{I}_h^{2h}\hat{V}_{i,t+h}^h] - \mathcal{I}_h^{2h}r_{i,t}^h.$$

Finally, we can derive the residual $r_t^h$ shown in Eq. 11. We fix $h = 0.02\,\mathrm{s}$ for multigrid refinement across all three case studies. The full procedure is summarized in Alg. 2.

### C.3.1 CONVERGENCE GUARANTEE

We demonstrate that our multigrid refinement guarantees convergence of the learned value $\hat{V}_i$ to the ground truth $V_i^*$ after sufficient training iterations. In our setting, each player's dynamics follow a

control-affine form

$$\dot{x}_i = \mathcal{A}_i(x_i) + \mathcal{B}_i(x_i)u_i$$

where $\mathcal{A}_i$ is continuously differentiable and $\mathcal{B}_i$ is smooth. Let $\hat{V}_i^k$ denote the learned value at epoch $k = 1, 2, ..., K$. Since the instantaneous cost $l_i$ only involves a quadratic control effort, given the optimal control $u_{-i}^*$ for Player $-i$, the Hamiltonian $\mathcal{H}_i$ for Player $i$ takes the form

$$\mathcal{H}_i = \max_{u_i \in \mathcal{U}_)} -\nabla_{x_i}\hat{V}_i(\mathcal{A}_i(x_i) + \mathcal{B}_i(x_i)u_i) - \nabla_{x_{-i}}\hat{V}_i(\mathcal{A}_{-i}(x_{-i}) + \mathcal{B}_{-i}(x_{-i})u_{-i}^*) + \partial_z \hat{V}_i u_i^T \mathcal{Q}_i u_i.$$

where $\mathcal{Q}_i$ is a symmetric and positive definite matrix. Our objective is to seek the $\hat{V}_i \leq 0$ when the state constraints $c_i(\mathbf{x}_i(s)) \leq 0$, $s \in [0, T]$ hold. Therefore, considering the structure of $\hat{V}_i$ in Eq. 13, we have $\partial_z \hat{V}_i = -1$ and yield

$$\mathcal{H}_i = \max_{u_i \in \mathcal{U}_i} -\nabla_{x_i}\hat{V}_i(\mathcal{A}_i + \mathcal{B}_i u_i) - \nabla_{x_{-i}}\hat{V}_i(\mathcal{A}_{-i} + \mathcal{B}_{-i}u_{-i}^*) - u_i^T \mathcal{Q}_i u_i.$$

Then we maximize $\mathcal{H}_i$ to derive the closed-form optimal control as the general formulation in our problem settings

$$u^* = -\frac{1}{2}\mathcal{Q}_i^{-1}\mathcal{B}_i^T[\nabla_{x_i}\hat{V}_i]^T \Rightarrow \mathcal{B}_i\nabla_{x_i}\hat{V}_i = -2[u^*]^T\mathcal{Q}_i$$

Therefore, when $\partial_t \hat{V}_i - \max_{u_i \in \mathcal{U}_i} H_i$ dominates in HJ PDEs, Eq. 5 reduces to the following form

$$\partial_t \hat{V}_i - H_i - u_i^T \mathcal{Q}_i u_i = 0 \Rightarrow \partial_t \hat{V}_i - H_i = -u_i^T \mathcal{Q}_i u_i.$$

Following Theorem 4.4 in Meng et al. (2024) and Theorem 3.1.4 in Jiang & Jiang (2017), given the closed-loop trajectory under control $u_i^{k+1}$, for each state-time pair $(\mathbf{x}_i, z_i, t)$, we have

$$\begin{aligned}
\hat{V}_i^{k+1} - \hat{V}_i^k &= \int_t^T \partial_t(\hat{V}_i^{k+1} - \hat{V}_i^k) + \nabla_{x_i}(\hat{V}_i^{k+1} - \hat{V}_i^k)(\mathcal{A}_i + \mathcal{B}_i u_i^{k+1}) \\
&\quad + \nabla_{x_{-i}}(\hat{V}_i^{k+1} - \hat{V}_i^k)(\mathcal{A}_{-i} + \mathcal{B}_{-i}u_{-i}^*)\, ds \\
&= \int_t^T -[u_i^{k+1}]^T \mathcal{Q}_i u_i^{k+1} + [u_i^k]^T \mathcal{Q}_i u_i^k - \mathcal{B}_i \nabla_{x_i}\hat{V}_i^k(u_i^{k+1} - u_i^k)\, ds \\
&= \int_t^T -[u_i^{k+1}]^T \mathcal{Q}_i u_i^{k+1} + [u_i^k]^T \mathcal{Q}_i u_i^k + 2[u^k]^T \mathcal{Q}_i(u_i^{k+1} - u_i^k)\, ds \\
&= \int_t^T -[u_i^{k+1}]^T \mathcal{Q}_i u_i^{k+1} - [u_i^k]^T \mathcal{Q}_i u_i^k + 2[u^k]^T \mathcal{Q}_i u_i^{k+1}\, ds \\
&= \int_t^T -||u_i^{k+1} - u_i^k||_{\mathcal{Q}_i}^2\, ds \\
&\leq 0
\end{aligned}$$

Eq. 2 shows that $\hat{V}_i$ is monotonic when $\int_t^T l_i\left(\mathbf{x}_i, u_i, u_{-i}^*\right) ds + g_i - z_i$ is dominated. After sufficient multigrid refinement steps, $\hat{V}_i^k$ converges to the ground truth $V_i^*$. In addition, since our refinement procedure follows the standard FAS, the convergence of each refinement step is guaranteed by classical multigrid theory (Trottenberg et al., 2001; Henson et al., 2003).

---

**Algorithm 2:** Multigrid Method for Value Approximation Refinement

---

**Input:** number of initial states $N$, time horizon $\mathcal{T} = [0, T]$, optimality operator $B[\cdot]$, restriction operator $\mathcal{I}_h^{2h}$, prolongation operator $\mathcal{I}_{2h}^h$, value $\hat{V}_i$ learned via epigraph-based PINN

**Initialize:** coarse-grid correction network $\varepsilon_{i,t}^{2h}, \forall t \in \mathcal{T}^{2h}$, fine-grid residual set $\mathcal{R}_i^h = \emptyset$, coarse-grid correction set $\mathcal{E}_i^{2h} = \emptyset$, coarse-grid time-steps $\mathcal{T}^{2h} = \{0, 2h, ..., T - 2h\}$, fine-grid time-steps $\mathcal{T}^h = \{0, h, ..., T - h\}$

1 **while** *until convergence* **do**
2      sampling $N$ initial states $(\mathbf{x}_i(0), z_i(0)) \in \mathcal{X}_{GT} \times \mathcal{Z}_i, \forall i = 1, 2$
3      generate rollouts using learned value $\hat{V}_i$ to collect training dataset $\mathcal{D}_M := \{(\mathbf{x}_i, z_i, t)^{(n)}\}_{n=1}^{N_M}$
     // Find-grid residual computation
4      **for** $t \in \mathcal{T}^h$ **do**
5          compute $B[\hat{V}_{i,t+h}^h]$ and fine-grid residuals $r_{i,t}^h = \hat{V}_{i,t}^h - B[\hat{V}_{i,t+h}^h]$ using $\mathcal{D}_M$
6          store $r_{i,t}^h$ in $\mathcal{R}_i^h$
7      **end**
     // Restriction and coarse-grid problem solving
8      **for** $t \in \mathcal{T}^{2h}$ **do**
9          compute $e_{i,t}^{2h}(\mathbf{x}_i, z_i) = B[\mathcal{I}_h^{2h}\hat{V}_{i,t+h}^h + e_{i,t+2h}^{2h}] - B[\mathcal{I}_h^{2h}\hat{V}_{i,t+h}^h] - \mathcal{I}_h^{2h}r_{i,t}^h$, where $e_{i,T}^{2h} = 0$
10          store $e_{i,t}^{2h}$ in $\mathcal{E}_i^{2h}$ and train coarse-grid correction network $\varepsilon_{i,t}^{2h}$ using $\mathcal{E}_i^{2h}$ and $\mathcal{D}_R$
11      **end**
     // Prolongation
12      **for** $t \in \mathcal{T}^h$ **do**
13          compute $e_{i,t}^h = \mathcal{I}_h^{2h}e_{i,t}^{2h}$ and update current value $\hat{V}_i$ to satisfy $\hat{V}_{i,t}^h \leftarrow \hat{V}_{i,t}^h + e_{i,t}^h$
14      **end**
     // Post Smoothing
15      **for** $t \in \mathcal{T}^h$ **do**
16          Fit fine-grid residuals $r_{i,t}^h = \hat{V}_{i,t}^h - B[\hat{V}_{i,t+h}^h]$ to 0
17      **end**
18 **end**

---

## C.4 HYPERPARAMETERS FOR ALL MODELS

**Network Architecture.** All methods employ fully connected neural networks with 3 hidden layers, where each layer involves 64 neurons. The activation functions follow the original configurations of each baseline: EMP, EPPO, and EHPINN use `tanh`, while EPIML uses `sin` (Sitzmann et al., 2020). To enhance training efficiency, all methods incorporate adaptive activation functions (Jagtap et al., 2020) and adopt the Adam optimizer with an adaptive learning rate[4], initialized at $1 \times 10^{-3}$ and decayed during training until reaching $1 \times 10^{-6}$. Detailed hyperparameter settings for all methods are provided in the following tables.

---

[4] EPPO uses a smaller initial learning rate of $1 \times 10^{-5}$ for improved convergence.

Table 4: Hyperparameters and Training Steps for EMP (Epigraph-based PINN)

| Parameter | Value |
|---|---|
| Network Architecture | MLP |
| Learning rate | Linear Decay $1 \times 10^{-3} \rightarrow 1 \times 10^{-6}$ |
| Neurons per Hidden Layer | 64 |
| Numbers of Hidden Layers | 3 |
| Optimizer | Adam |
| Activation Function | `tanh` |
| **Case 1** | |
| Number of Pre-training Steps | $20,000$ |
| Number of Training Steps | $300,000$ |
| $(C_1, C_2, C_3)$ | $(1500, 0.65, 0.45)$ |
| **Case 1 (EMP without multigrid)** | |
| Number of Pre-training Steps | $20,000$ |
| Number of Training Steps | $300,000$ |
| **Case 2** | |
| Number of Pre-training Steps | $20,000$ |
| Number of Training Steps | $300,000$ |
| $(C_1, C_2, C_3)$ | $(6000, 2, 0.65)$ |
| **Case 2 (EMP without multigrid)** | |
| Number of Pre-training Steps | $20,000$ |
| Number of Training Steps | $300,000$ |
| **Case 3** | |
| Number of Pre-training Steps | $20,000$ |
| Number of Training Steps | $300,000$ |
| $(C_1, C_2, C_3)$ | $(7000, 1.25, 0.125)$ |
| **Case 3 (EMP without multigrid)** | |
| Number of Pre-training Steps | $20,000$ |
| Number of Training Steps | $440,000$ |

Table 5: Hyperparameters and Training Steps for EMP (Multigrid Refinement)

| Parameter | Value |
|---|---|
| Network Architecture | MLP |
| Learning rate | Linear Decay $1 \times 10^{-3} \rightarrow 1 \times 10^{-6}$ |
| Neurons per Hidden Layer | 64 |
| Numbers of Hidden Layers | 3 |
| Optimizer | Adam |
| Activation Function | `tanh` |
| **Case 1** | |
| Number of Training Steps | $10,000$ Gradient Descent per Epoch over 15 Epochs |
| **Case 1 (EMP without rollout)** | |
| Number of Training Steps | $10,000$ Gradient Descent per Epoch over 30 Epochs |
| **Case 2** | |
| Number of Training Steps | $10,000$ Gradient Descent per Epoch over 5 Epochs |
| **Case 2 (EMP without rollout)** | |
| Number of Training Steps | $10,000$ Gradient Descent per Epoch over 30 Epochs |
| **Case 3** | |
| Number of Training Steps | $10,000$ Gradient Descent per Epoch over 80 Epochs |
| **Case 3 (EMP without rollout)** | |
| Number of Training Steps | $10,000$ Gradient Descent per Epoch over 80 Epochs |

Table 6: Hyperparameters and Training Steps for EPPO

| Parameter | Value |
|---|---|
| Network Architecture | MLP |
| Actor Learning rate | Linear Decay $1 \times 10^{-5} \to 1 \times 10^{-6}$ |
| Critic Learning rate | Linear Decay $1 \times 10^{-5} \to 1 \times 10^{-6}$ |
| Units per Hidden Layer | 64 |
| Numbers of Hidden Layers | 3 |
| Optimizer | Adam |
| Discount factor $\gamma$ | 0.99 |
| Actor Net Activation Function | `tanh` |
| Critic Net Activation Function | `tanh` |
| Clip Ratio | 0.2 |
| Seed | 0,1,2,3,4 |
| **Case 1** | |
| Number of Training Steps | $1,150,000$ |
| **Case 2** | |
| Number of Training Steps | $1,150,000$ |
| **Case 3** | |
| Number of Training Steps | $1,800,000$ |

Table 7: Hyperparameters and Training Steps for EHPINN

| Parameter | Value |
|---|---|
| Network Architecture | MLP |
| Learning rate | Linear Decay $1 \times 10^{-3} \to 1 \times 10^{-6}$ |
| Neurons per Hidden Layer | 64 |
| Numbers of Hidden Layers | 3 |
| Optimizer | Adam |
| Activation Function | `tanh` |
| Seed | 0,1,2,3,4 |
| **Case 1** | |
| Number of Pre-training Steps | $20,000$ |
| Number of Training Steps | $200,000$ |
| **Case 2** | |
| Number of Pre-training Steps | $20,000$ |
| Number of Training Steps | $200,000$ |
| **Case 3** | |
| Number of Pre-training Steps | $20,000$ |
| Number of Training Steps | $200,000$ |

Table 8: Hyperparameters and Training Steps for EPIML

| Parameter | Value |
|---|---|
| Network Architecture | MLP |
| Learning rate | Linear Decay $1\times10^{-3} \to 1\times10^{-6}$ |
| Neurons per Hidden Layer | 64 |
| Numbers of Hidden Layers | 3 |
| Optimizer | Adam |
| Activation Function | sin |
| Seed | 0,1,2,3,4 |
| **Case 1** | |
| Number of Pre-training Steps | $20,000$ |
| Number of Training Steps | $330,000$ |
| **Case 2** | |
| Number of Pre-training Steps | $20,000$ |
| Number of Training Steps | $330,000$ |
| **Case 3** | |
| Number of Pre-training Steps | $20,000$ |
| Number of Training Steps | $400,000$ |

Table 9: Hyperparameters and Training Steps for MADDPG

| Parameter | Value |
|---|---|
| Network Architecture | MLP |
| Actor Learning rate | Linear Decay $1\times10^{-5} \to 1\times10^{-6}$ |
| Critic Learning rate | Linear Decay $1\times10^{-5} \to 1\times10^{-6}$ |
| Units per Hidden Layer | 64 |
| Numbers of Hidden Layers | 3 |
| Optimizer | Adam |
| Discount factor $\gamma$ | 0.99 |
| Actor Net Activation Function | tanh |
| Critic Net Activation Function | tanh |
| Seed | 0,1,2,3,4 |
| **Case 1** | |
| Number of Training Steps | $1,200,000$ |
| **Case 2** | |
| Number of Training Steps | $1,400,000$ |
| **Case 3** | |
| Number of Training Steps | $2,500,000$ |

