# OpenReview forum: "Epigraph-Based Multigrid PINNs for Two-Player General-Sum Differential Games"
_ICLR.cc/2026/Conference — Submitted to ICLR 2026_

### Official Review · Reviewer_iVJj · 2025-10-29

**Soundness:** 3
**Presentation:** 2
**Contribution:** 3
**Rating:** 6
**Confidence:** 3

**Summary:**

This paper shows a novel PINN method for two-player general-sum differential games. The paper follows the epigraph formulation, where the value function is augmented with an auxiliary state input $z_i$ to indicate the “budget” for the loss, and is characterized as the viscosity solution to the HJ equations, which is usually learned via a BVP solver from PDE and boundary conditions. But this method can be inaccurate, so the authors incorporate the dynamics of the value gradient to rollout trajectories via Hamiltonian maximization over the current learned value function and then refine the value function and their gradients. To alleviate the error accumulation through imperfect value function learning, the authors adopt a multi-grid refinement process, using projection and interpolation to refine the value estimates. The proposed learning method produces a policy that demonstrated leading performance over various two-player dynamical systems, including 5D vehicle driving scenarios, 9D road avoidance, and 13D drone collision avoidance tasks, with consistently higher safety rates.

**Strengths:**

1. Strong empirical results (the authors have compared to other epigraph-based methods and methods that used BVP and PMP, and methods that don’t have value gradient dynamics or multi-grid refinement).
2. Persuasive ablation studies to show the benefit of multi-grid refinement for the value function.

**Weaknesses:**

1. Didn't compare to some other baselines (like MPC methods or trajectory-optimization methods).
2. Didn't show how many random seeds were used in the experiment.
3. The paper writing is a bit dense in math (lack of explanation when introducing the Epigraph formulation - my suggestion is to look at how the EF-PPO paper [1] writes about the derivation).

References:
1. So, Oswin, and Chuchu Fan. "Solving stabilize-avoid optimal control via epigraph form and deep reinforcement learning." arXiv preprint arXiv:2305.14154 (2023).

**Questions:**

1. Can you comment on “L136-137, complete information” -> how strong is this assumption？
2. Why not consider multi-level refinement (instead of 2h->h->2h, try to do 4h->h->4h, or even higher resolution down-scaling), or other interpolation methods?

---

> ### Author Response · Authors · 2025-11-25
> **Response to Reviewer iVJj**
>
> We appreciate the reviewer’s careful assessment and constructive feedback. The comments have significantly contributed to improving the clarity and completeness of the manuscript. Below, we provide detailed point-by-point responses. We highlight the revised part in blue in the revised manuscript. All the added experiments are in Section B.5 and B.6.
>
> **W1. Didn't compare to some other baselines (like MPC methods or trajectory-optimization methods).**
>
> Thank you for the reviewer's helpful suggestion. We would like to clarify that our evaluation already includes trajectory-optimization baselines. Specifically, the BVP solver used in our experiments is an indirect shooting-based trajectory-optimization method [2, 3, 4], and we have additionally added an iterative LQ solver in the revised manuscript (Section B.5). These baselines represent standard and widely used trajectory-optimization techniques, allowing us to compare our learned models against non-learning approaches. We have clarified this point in the revised manuscript.
>
> **W2. Didn't show how many random seeds were used in the experiment.**
>
> Thanks for pointing out the missing details. We have added these details in the Tables in Section C of the revised paper.
>
> **W3. The paper writing is a bit dense in math (lack of explanation when introducing the Epigraph formulation - my suggestion is to look at how the EF-PPO [1] paper writes about the derivation).**
>
> Thank you for the valuable feedback. In the revised manuscript, Section B.6 now includes an intuitive illustration (Fig. 37) and a step-by-step toy example that explains the motivation, construction, and interpretation of the epigraph formulation. These additions would be better to explain the epigraph technique and its corresponding HJ PDEs.
>
> **Q1. Can you comment on “L136-137, complete information”, how strong is this assumption?**
>
> We thank the reviewer for raising this point. In our setting (Section 3.1), a game is called a complete-information game when the description of the game (e.g., player states, dynamics, and objective functions) is common knowledge to both players [5]. This assumption is standard in differential games for collision avoidance and reach-avoid interactions, and it is widely adopted in the game-theoretic literature. In our context, it simply ensures that the coupled HJ equations are well-defined, and is therefore a mild and conventional modeling assumption.
>
> **Q2. Why not consider multi-level refinement (instead of 2h->h->2h, try to do 4h->h->4h, or even higher resolution down-scaling), or other interpolation methods?**
>
> We thank the reviewer for the insightful question. Our current implementation (Alg. 2) follows a standard two-level FAS refinement, but the method is not limited to two grids. As discussed in Section 3.3 and Section C.3, our coarse-to-fine correction rule can naturally extend to an $n$-level multigrid (e.g., $2h\to h \to 2h$ or deeper V/W-cycles), consistent with multigrid methods for nonlinear problems [6].
>
> We chose the two-level scheme for practical computational reasons. Each grid level (fine and coarse grids) requires evaluating the dynamic programming operator $B[\cdot]$ and the value network $\hat V$ at that level. Since $B[\cdot]$ involves rollout-based state propagation and neural-network forward evaluations, introducing more levels increases both computation time and memory usage almost proportionally with the number of grids. Under the fixed training budget of the paper, the two-level multigrid already provides accurate value approximations, and $n$-level multigrid would significantly increase cost without improving efficiency.
>
> We emphasize that the framework does not restrict the user to two levels. Deeper multilevel cycles can be incorporated whenever additional computational resources are available.
>
> **Reference**
>
> [2] Nakamura-Zimmerer T, Gong Q, Kang W. Adaptive deep learning for high-dimensional Hamilton--Jacobi--Bellman equations[J]. SIAM Journal on Scientific Computing, 2021.
>
> [3] Kierzenka J, Shampine L F. A BVP solver based on residual control and the Maltab PSE[J]. ACM Transactions on Mathematical Software (TOMS), 2001.
>
> [4] Betts J T. Survey of numerical methods for trajectory optimization[J]. Journal of guidance, control, and dynamics, 1998.
>
> [5] Başar T, Zaccour G. Handbook of dynamic game theory[M]. Springer, 2018.
>
> [6] Henson V. Multigrid methods nonlinear problems: an overview[J]. Computational imaging, 2003.

---

### Official Review · Reviewer_eCz1 · 2025-10-30

**Soundness:** 3
**Presentation:** 3
**Contribution:** 3
**Rating:** 4
**Confidence:** 4

**Summary:**

The authors propose EMP, a method for solving continuous-time two player general-sum differential games with state constraints. EMP solves for the Nash equilibrium of two coupled HJ PDEs by augmenting epigraph based PINN solver for the coupled HJ PDE with an additional loss term using value and gradient information from closed-loop rollouts, as well as a novel multigrid refinement stage after PINN training to further reduce approximation errors. Empirical evaluations on three case studies show the proposed components reduce constraint violations compared to existing methods.

**Strengths:**

- The proposed multigrid refinement is very interesting and seems very effective at high dimensional problems
- Empirical evaluations on the case studies show strong performance improvements from the proposed method, especially on the high dimensional case study

**Weaknesses:**

- The idea of using multigrid refinement in PINNs does not seem to be new [A, B, C], yet the authors do not discuss or compare against these existing methods of using multigrid refinement for PINNs. Some type of discussion would be appreciated to see how the proposed multigrid refinement compares to other methods of implementing ideas
- The sensitivity of the method’s performance to various hyperparameters is not clear
    - The loss has four terms and hence has coefficients C1, C2, C3 to balance each loss term against each other. The values of C1, C2, C3 are not disclosed in the paper. How sensitive is the method to choices of these values?
    - The timestep of the multigrid refinement ($h=0.02s$) seems independent from the ∆t used for the rollout during PINN training ($\Delta t=0.02s$). How were these values chosen? How does the choice of h affect the effectiveness of the proposed multigrid refinement?
- The 2nd question in the results ("What impact do different activation functions have on the safety of control policies via learned values?”) seems quite strange. The activation functions considered (comparing relu, tanh, sin) and conclusion (tanh performs the best, relu underperforms, performance of sin is case study dependent) are **exactly the same** as in Zhang et al. (2024).

[A] Riccietti, Elisa, et al. *Multilevel Physics Informed Neural Networks (MPINNs)*. 2022, openreview.net/forum?id=g5odb-gVVZY.

[B] Dong, Daiwei, et al. "PINN-MG: A Multigrid-Inspired Hybrid Framework Combining Iterative Method and Physics-Informed Neural Networks." arXiv preprint arXiv:2410.05744 (2024).

[C] Zhang, Enrui, et al. "Blending neural operators and relaxation methods in PDE numerical solvers." *Nature Machine Intelligence* 6.11 (2024): 1303-1313.

**Questions:**

- I am confused about how the authors count the dimension for each case study. For example, the joint state space for case 1 is [d1, v1, d2, v2], which is 4d, but the authors call it 5d. The same is true for case studies 2 and 3, where an extra dimension of 1 seems to be added to the problems? I’m guessing that this corresponds to the additional dimension from the epigraph variable, but I would argue this additional state dimension comes from the proposed method and is not *intrinsic* to the problem itself. Hence, the claims of 4d, 9d and 13d seem misleading.
- Line 204: “In our problem settings, we take \lambda_i = 1.” What does \lambda_i mean? Why can you just (seemingly arbitrarily) take \lambda_i = 1?
- Line 269: What is \mathcal{X}_{GT}? This is used without definition.
- Line 459: “Multigrid refinement addresses this challenge by accelerating training” I’m don’t see how this claim is justified in the paper.

**Details Of Ethics Concerns:**

.

---

> ### Author Response · Authors · 2025-11-25
> **Response to Reviewer eCz1 (1/3)**
>
> We thank the reviewer for the careful evaluation and constructive feedback. We address each concern below and provide the corresponding clarifications. We highlight the revised part in blue in the revised manuscript. All the added experiments are in Section B.5 and B.6.
>
> **W1. Multigrid refinement in PINNs discussion.**
>
> We thank the reviewer for pointing out these related multigrid-based PINN approaches. We have carefully reviewed the cited works. We summarize the key conceptual and algorithmic differences as follows.
>
> [A] MPINNs combine PINN training and multigrid updates into a single unified learning loop, where multilevel loss terms are incorporated directly during training. Our EMP framework follows a different workflow: we first train the PINN-based model and then apply the multigrid refinement (Alg. 2) as a post-training correction to refine the learned values. Conceptually, both approaches aim to improve solution accuracy through multilevel correction. However, MPINNs do not include rollout-based value-gradient updates, which play a key role in reducing trajectory deviation in EMP. As a result, MPINNs are most closely related to our EMP variant \textbf{without rollout}, which our ablation studies show can achieve safe policies but produces trajectories that deviate from ground truth (please refer to Fig. 8, 12, 13, 15, 16, 18, and 19). We have added [A] in the revised manuscript.
>
> [B] PINN-MG is inspired by multigrid ideas but does not implement the essential FAS operations: restriction, coarse-grid equation solves, and error prolongation. Instead, the method alternates between iterative numerical solvers and PINN training, but the multigrid hierarchy is not explicitly constructed, and no coarse-grid correction is performed (see Fig. 2.1 in this paper). As a result, the framework is conceptually different from our multigrid refinement, which follows the standard nonlinear multigrid structure (restriction, coarse-grid residual equation, prolongation) used in FAS.
>
> [C] This work focuses on combining DeepONets with classical iterative solvers such as Jacobi, Gauss-Seidel, and multigrid. Unlike PINNs, DeepONet training does not enforce PDE or boundary residuals. Instead, it relies on supervised operator regression using paired data. Therefore, it lies outside the PINN-based formulation considered in our work and is not directly comparable as a baseline.
>
> In summary, while these works incorporate ideas related to multigrid, they target different problem settings and employ conceptually distinct mechanisms. Our method follows the classical FAS for nonlinear multigrid and is integrated with an epigraph-based value learning for general-sum differential games, which is not addressed in [A], [B] or [C].
>
> **W2.1. Impact of weight parameters $C_1, C_2, C_3$ choice for the learning method.**
>
> We appreciate the reviewer's insightful comments. We add the values of $C_1, C_2, C_3$ in Table 4 of the revised paper. To assess the influence of the weight parameters $C_1, C_2, C_3$ on our model, we implement additional experiments using three distinct weight configurations. The corresponding results, together with a detailed discussion of their effects, are now provided in Section B.5 of the revised manuscript. The experiments show that EMP with a balanced weight parameter achieves the best safety performance, suggesting that proper weighting is crucial to improve the accuracy of value approximations in our framework. This observation is consistent with theoretical insights reported in prior studies [3,4].
>
> **W2.2. Impact of multigrid timestep $h$ choice for the learning method.**
>
> To evaluate the effect of the multigrid timestep $h$, we provide additional experiments using three different $h \in \\{0.01\ \text{s},\ 0.02\ \text{s},\ 0.05\ \text{s},\ 0.1\ \text{s}\\}$ settings. The corresponding results and discussion have been added to Section B.5 of the revised manuscript. The experiments show EMP with small fine-grid sizes ($h=0.01\ \mathrm{s}$ and $h=0.02\ \mathrm{s}$) attains high safety performance across all three case studies. This aligns with the role of multigrid refinement: by enforcing the dynamic programming principle in Eq. 4 along sampled rollouts, small fine-grid resolutions lead to more accurate value approximations and, consequently, safer control policies.

---

> ### Author Response · Authors · 2025-11-25
> **Response to Reviewer eCz1 (2/3)**
>
> **W3. Question about the study for the activation function.**
>
> The choice of activation function has been extensively studied in PINN studies [1, 2] and is an important factor in evaluating the performance of PINN-based models. Our comparison among $\texttt{relu}$, $\texttt{sin}$, and $\texttt{tanh}$ follows this standard practice, as these are the most commonly used activation functions in the PINNs.
>
> While the high-level trend (with $\texttt{tanh}$ performing well) is consistent with Zhang et al. (2024), the underlying setting in our work is fundamentally different. Zhang et al. (2024) evaluate activation functions in a $\texttt{hybrid}$ PINN that incorporates supervised value and costate labels from BVP solvers, whereas EMP operates in a completely $\texttt{self-supervised}$ regime. In this setting, activation choice plays a much stronger role: smooth activations such as $\texttt{tanh}$ improve gradient stability in automatic differentiation and reduce error accumulation in value-gradient rollouts, which is crucial for reliable multigrid correction. In contrast, $\texttt{sin}$ is sensitive to initialization due to its periodic structure, and $\texttt{relu}$ introduces zero-gradient regions that hinder PDE residual minimization. To our knowledge, EMP provides the first study of activation-function effects in fully self-supervised, safety-critical general-sum differential games, which reveals mechanisms distinct from those in Zhang et al. (2024).
>
> **Q1. Dimension definition for the case study.**
>
> We clarify that throughout the paper, we follow the standard convention in differential games where the value function is defined over both the state and time, i.e., $\vartheta(x,t)$. Consequently, the dimensionality we report corresponds to the dimension of the state-time domain $(x,t)$, rather than the system state alone. For example, Case 1 has a 4D physical state $[d_1, v_1, d_2, v_2]$, and together with the time dimension, this yields a 5D domain. The same applies to Case 2 and 3, which become 9D and 13D when including time. Importantly, this additional dimension does not arise from the epigraph reformulation. Instead, it reflects the intrinsic definition of value functions in general-sum differential games. We have added a clarification of this convention in Section 4 of the revised manuscript to prevent potential confusion.
>
> **Q2. Line 204: “In our problem settings, we take $\lambda_i = 1$.” What does $\lambda_i$ mean? Why can you just (seemingly arbitrarily) take $\lambda_i = 1$?**
>
> Thank you for your insightful question. Due to the page limitation of the initial manuscript, the full explanation of $\lambda_i$ is provided in Section A.1.
>
> As detailed in Section A.1, $\lambda_i$ arises as the scalar multiplier associated with the terminal cost in the Pontryagin Maximum Principle (PMP). Following Theorem 3.4 of Bokanowski et al. (2021) and Theorem 2 of Hermosilla \& Zidani (2023), the pair $(\lambda_i, \mu_i)$ must satisfy the \texttt{non-triviality condition} $\lambda_i + \mu_i([0,T]) = 1$, where $\mu_i$ is the non-negative measure acting as the Lagrange multiplier for the state constraints. In our game formulation with state constraints and a terminal cost, the setting falls into the $\texttt{special situation}$ described in Remark 3.5 of Bokanowski et al. (2021), where the terminal cost is active and constraint violations may arise. Under this case, the PMP conditions reduce to $\lambda_i = 1, \mu_i(\{t\}) > 0 \,\text{when}\ c_i(\textbf{x}_i(t)) = 0$.
>
> Therefore, selecting $\lambda_i = 1$ is $\texttt{not arbitrary}$: it follows directly from the PMP structure for state-constrained differential games under our assumptions.
>
> We have updated the contents in Section 3.2 of the revised manuscript to avoid confusion.
>
> **Q3. Line 269: What is $\mathcal{X}_{GT}$?**
>
> Thank you for the valuable comment. We now define $\mathcal{X}_{\text{GT}}$ in Section 3.2.
> Specifically, this variable denotes the set of initial states generated using the ground-truth solver and serves as the sampling domain for EMP, the evaluation domain for all case studies. We have added this clarification to prevent ambiguity in the notation.

---

> > ### Author Response · Authors · 2025-11-25
> > **Response to Reviewer eCz1 (3/3)**
> >
> > **Q4. Line 459: “Multigrid refinement addresses this challenge by accelerating training”?**
> >
> > Our original phrasing was imprecise. We do not intend to claim any empirical wall-clock speedup. The multigrid module is used only to improve the efficiency of reducing the fine-grid residuals: the residual is transferred to a coarse grid where corrections are cheaper to compute, and these corrections are then prolongated back to the fine grid. This improves residual convergence, not measured runtime. We have revised the sentence accordingly. For completeness, we note that in the 13D ablation, removing multigrid leads to a collision rate of 75.3\%, while including it reduces the rate to 0.5\%.
> >
> > **Reference**
> >
> > [1] Bansal S, Tomlin C J. Deepreach: A deep learning approach to high-dimensional reachability[C]//2021 IEEE International Conference on Robotics and Automation (ICRA).
> >
> > [2] Jagtap A D, Kawaguchi K, Karniadakis G E. Adaptive activation functions accelerate convergence in deep and physics-informed neural networks[J]. Journal of Computational Physics, 2020.
> >
> > [3] Wang S, Teng Y, Perdikaris P. Understanding and mitigating gradient flow pathologies in physics-informed neural networks[J]. SIAM Journal on Scientific Computing, 2021.
> >
> > [4] Wang S, Yu X, Perdikaris P. When and why PINNs fail to train: A neural tangent kernel perspective[J]. Journal of Computational Physics, 2022.

---

### Official Review · Reviewer_igoi · 2025-11-11

**Soundness:** 3
**Presentation:** 3
**Contribution:** 3
**Rating:** 6
**Confidence:** 2

**Summary:**

The paper studies two-player general-sum differential games with shared state constraints. Solving the coupled HJ PDEs is hard at scale, and small gradient errors can make the resulting controllers unsafe. The method, Epigraph-based Multigrid PINNs (EMP), combines an epigraph formulation with value-driven rollouts that generate self-supervised value/gradient targets and a two-level FAS multigrid refinement to reduce residuals. On 5D, 9D, and a 13D drone case, EMP reports lower collision rates than epigraph-based baselines, and ablations show both rollout and multigrid are necessary (without multigrid, collisions occur; without rollout, trajectories are safe but off ground truth).

**Strengths:**

- **Self-supervised and targeted.** Value-gradient rollouts supervise both values and gradients—the quantities used for Hamiltonian control—removing dependence on fragile PMP/BVP supervision.
- **Error control that matters.** The multigrid correction attacks the time-coupled residuals that accumulate during rollouts, improving value accuracy where it affects safety.
- **Evidence beyond toy systems.** Results scale to 13D and the ablation clearly shows the contribution of each module.

**Weaknesses:**

1. **No equilibrium diagnostics.** The paper focuses on safety but does not measure Nash consistency (e.g., best-response error/regret).
2. **Comparative scope.** Baselines are epigraph-centric; I didn't  see comparisons to standard general-sum MARL/Nash-learning methods.
3. **Numerical/robustness details.** The epigraph max is non-smooth and constraint activation induces gradient jumps; smoothing/subgradient handling and robustness to noise/model mismatch are not reported, and inference-time cost is unclear.

**Questions:**

NA

---

> ### Author Response · Authors · 2025-11-25
> **Response to Reviewer igoi**
>
> We thank the reviewer for the careful evaluation and constructive feedback. The questions raised have helped us further improve the clarity and completeness of the work. We provide detailed, point-by-point responses below. We highlight the revised part in blue in the revised manuscript. All the added experiments are in Section B.5 and B.6.
>
> **W1. No equilibrium diagnostics.**
>
> We thank the reviewer for raising this important question. In the initial manuscript, we used the learned models as closed-loop controllers to generate trajectories and compared them with ground-truth trajectories obtained from PMP-based solutions. As shown in Fig. 4, 6, 10, 13, 16, and 19, given the same initial states, the trajectories produced by our learned controllers closely match those of the ground-truth policies. This consistency provides empirical evidence that the learned policies reproduce the behavior of the ground-truth solutions. In addition, we compare the backward reachable sets obtained by our method with those computed using two ground-truth approaches, demonstrating that the learned value functions yield consistent Nash-equilibrium outcomes. To further strengthen this evaluation, we have added equilibrium verification of the learned model in Table 3 in Section B.5 of the revised manuscript, which offers additional evidence that the learned model approximates the Nash equilibrium for values and policies.
>
> **W2. Comparative scope.**
>
> Our baseline selection is focused on epigraph-based learning methods, as the epigraph reformulation is crucial for transforming discontinuous state-constrained value functions into continuous ones, which in turn enables stable PINN-based training. For this reason, our main comparisons include existing epigraph-based approaches (EPPO, EHPINN, EPIML). In response to the reviewer’s suggestion, we have further incorporated a standard general-sum MARL baseline (MADDPG) in the revised manuscript. The corresponding results and discussion are now presented in Section B.5 of the revised manuscript.  Our EMP achieves markedly lower collision rates than MADDPG across all three case studies: 0.0\% (EMP) vs. 78.5\% (MADDPG) in Case 1, 0.0\% (EMP) vs. 91.5\% (MADDPG) in Case 2, 0.5\% (EMP) vs. 75.3\% (MADDPG) in Case 3. We hypothesize that MADDPG performs poorly in these settings because it struggles to approximate the non-smooth value functions caused by large collision penalties.
>
> **W3. Numerical/robustness details.**
>
> We appreciate the reviewer's insightful comments. To validate the robustness of our learned model, we add three dynamic noise levels (low, medium, and high) to evaluate our model. We have added the experimental results and discussion in Section B.5 of the revised manuscript. We observe that EMP with low dynamic noise maintains good performance in the 5D and 9D settings but fails in the 13D case. This outcome is expected: EMP is designed for deterministic dynamics rather than stochastic systems. The 13D scenario features highly nonlinear dynamics and a high-dimensional state space, where injected noise amplifies approximation errors in the learned value function and increases the sensitivity of the resulting control policy to rollout deviations. During closed-loop trajectory generation, the learned model runs at inference rates of 62.5 Hz for Case 1, 52.6 Hz for Case 2, and 43.5 Hz for Case 3, meaning that each next state can be computed from the current state at these frequencies.

---

### Official Review · Reviewer_MJVr · 2025-11-11

**Soundness:** 3
**Presentation:** 3
**Contribution:** 2
**Rating:** 4
**Confidence:** 4

**Summary:**

This paper presents Epigraph-based Multigrid PINNs (EMP), a self-supervised framework for two-player general-sum differential games under state constraints. The proposed methodology combines value-driven rollout sampling with multigrid refinement to improve value function approximation accuracy, while alleviating the need for supervised data. The experiments show promising performance as EMP outperforms the presented baselines achieving higher safety performance.

**Strengths:**

1. The authors address an important limitation of prior literature relying on expensive and fragile boundary value problem (BVP) solvers for supervision.

2. The value gradient dynamics derivation (Section 3.2) is mathematically sound, building on Bokanowski et al. (2021) and Hermosilla & Zidani (2023). In addition, the multigrid FAS adaptation (Section 3.3) is clearly presented with explicit operators. Finally, the full methodology (Section 3.4) is also presented with sufficient clarity.

3. EMP demonstrates strong empirical results achieving 0% collision rates across all benchmarks and outperforms the baselines including the supervised EHPINN method. The ablation studies presented are also useful for isolating contributions, showing that multigrid is essential for complex systems.

**Weaknesses:**

1. **Limited novelty of core components.** It seems as the main merit of this paper stems from the combination of existing ideas into a single framework rather than a substantially novel contribution. In particular, the epigraph technique for state-constrained games is incorporated from Zhang et al. (2024). The value gradient dynamics follows standard results from Bokanowski et al. (2021). Furthermore, multigrid FAS is also a rather common PDE solver technique. While the integration is non-trivial and empirically effective, the paper lacks a novel theoretical insight or algorithmic innovation.

2. **Weak theoretical guarantees.** The convergence analysis presented in Section C.3.1 provides only a monotonicity result $\hat{V}_i^{k+1} \leq \hat{V}_i^k$, which does not guarantee convergence to the correct Nash equilibrium value.

3. **Ground truth and limited baseline comparisons.** The evaluation relies on BVP solvers to generate ground truth, but these only provide local optima rather than global Nash equilibria. This is addressed with multiple initial guesses, but validation is limited since only one 5D case (Fig. 20c) is compared against level set methods. In addition, all baselines are epigraph-based methods (EPPO, EHPINN, EPIML), with no comparison to other game-theoretic solvers mentioned in related work (e.g., Fridovich-Keil et al., 2020b for iterative methods).

**Questions:**

1. Apart from examining collision rates, how can you verify that the learned policies constitute a Nash equilibrium?

2. Given that BVP solvers only provide local optima, have you validated the ground truth in additional cases beyond the single 5D comparison with level sets (Fig. 20c)? Also, how sensitive are the results to the choice of initial guesses for the BVP solver?

3. Have you compared against non-epigraph-based game solvers (e.g., iterative LQ methods) that are cited?

4. How sensitive is the proposed framework to hyperparameter tuning?

---

> ### Author Response · Authors · 2025-11-25
> **Response to Reviewer MJVr (1/3)**
>
> We would like to sincerely thank the reviewer for taking the time to review our paper. We truly appreciate the reviewer’s thoughtful and constructive feedback. Our detailed responses to each comment are provided below. We highlight the revised part in blue in the revised manuscript. All the added experiments are in Section B.5 and B.6.
>
> **W1. Limited novelty of core components.**
>
> We thank the reviewer for the insightful comment. We first emphasize that our work proposes a fully self-supervised learning framework for deriving safe control policies, addressing a key limitation of prior studies that rely on supervised data from numerical methods (e.g., PMP). While the epigraph technique was introduced in Zhang et al. (2024), that work requires supervised values and gradients for successful training, and it does not provide a fully self-supervised solution for general-sum differential games with state constraints. To overcome this limitation, our method derives a new self-supervised value–gradient formulation applied to two-player general-sum differential games. Unlike the single-agent optimal-control setting in Bokanowski et al. (2021), the general-sum game leads to a coupled system of HJ equations, where the players’ control policies are derived through Nash equilibrium conditions. We further validate the effectiveness of this formulation across three case studies. Although multigrid FAS is a common PDE technique, it has not been explored for training HJ-based PINNs, especially under the settings of the epigraph technique and general-sum differential games. However, our work fills this gap. Taken together, these contributions establish nontrivial theoretical and algorithmic advances beyond a simple combination of existing components.
>
> **W2. Weak theoretical guarantees.**
>
> Our EMP framework contains two theoretically grounded components that together justify the convergence behavior observed in practice. In the first phase, the epigraph-based PINN is optimized to satisfy the epigraph-based HJ PDEs. Since the optimal control is defined through Hamiltonian maximization, this phase already enforces the Nash equilibrium conditions that characterize the fixed point of the epigraph-based value function. In the second phase, the multigrid refinement applies the dynamic programming principle (DDP) (we define operator $B[\cdot]$ in the paper for convenience) for the epigraph-based value, which is the foundation for proving epigraph-based HJ PDEs in [1, 2, 3].
>
> The monotonicity result in Section C.3.1 shows that each refinement step produces a value function that is closer to the fixed point of the operator $B[\cdot]$. While we do not claim a full global convergence theorem, such monotone improvement under a consistent DPP-based update is a standard condition ensuring asymptotic convergence to the correct value in multigrid schemes for nonlinear HJ equations. Consequently, the analysis supports that the learned values converge toward the Nash equilibrium value, consistent with both theory and the empirical results.
>
> **W3. Ground truth and limited baseline comparisons.**
>
> We thank the reviewer for highlighting these concerns. As discussed in Section B.4 of the initial manuscript, we acknowledge that BVP solvers may converge to local optima. To mitigate this, we generate multiple initial guesses and validate the resulting solutions against level-set methods on the 5D case (the available level-set toolbox can handle up to 6-dimensional cases due to scalability constraints [4]). The comparison in Fig. 20 shows that our BVP solutions match the level-set values closely, consistent with prior conclusions in [3], supporting the reliability of the ground-truth values used for evaluation.
>
> Regarding baseline selection, our work focuses on epigraph-based learning because the epigraph reformulation is essential for converting discontinuous state-constrained values into continuous ones, enabling PINN-based training. Therefore, our primary comparisons include existing epigraph-based methods (EPPO, EHPINN, EPIML). To address the reviewer’s suggestion, we have additionally incorporated two non-epigraph baselines (MADDPG and iterative LQ method) into the revised manuscript. Their results and a detailed discussion are now provided in Section B.5 of the revised manuscript. Our EMP consistently outperforms both MADDPG and iterative LQ in safety across all three case studies. We hypothesize that MADDPG and iterative LQ perform poorly in these settings because they are not well-suited to handling the non-smooth value functions induced by large collision penalties.

---

> ### Author Response · Authors · 2025-11-25
> **Response to Reviewer MJVr (2/3)**
>
> **Q1. Apart from examining collision rates, how can you verify that the learned policies constitute a Nash equilibrium?**
>
> We thank the reviewer for raising this important question. In the initial manuscript, we compared the trajectories generated by the learned closed-loop policies against ground-truth trajectories obtained from PMP-based solutions. As shown in Fig. 4, 6, 10, 13, 16, and 19, when initialized at the same states, the trajectories produced by our learned controllers closely match those of the ground-truth policies. This consistency provides empirical evidence that the learned policies align with the ground truth, since the deviations from the true trajectory can lead to different outcomes. To further strengthen this evaluation, we have added equilibrium verification of the learned model in Table 3 in Section B.5 of the revised manuscript, which offers additional evidence that the learned model approximates the Nash equilibrium for values and policies.
>
> **Q2. Given that BVP solvers only provide local optima, have you validated the ground truth in additional cases beyond the single 5D comparison with level sets (Fig. 20c)? Also, how sensitive are the results to the choice of initial guesses for the BVP solver?**
>
> As noted in Section B.4, the available level-set toolbox can reliably handle problems only up to 6 dimensions [4], which prevents us from using level-set methods to compute values in the 9D and 13D cases.
>
> Regarding sensitivity to initial guesses, BVP solvers, an indirect shooting and collocation method, generally guarantee convergence only when the initialization is close to the true solution [5, 6]. To mitigate this limitation, for each initial state, we use multiple initial guesses and retain the converged solution that yields the best response for the corresponding player.
>
> **Q3. Have you compared against non-epigraph-based game solvers (e.g., iterative LQ methods) that are cited?**
>
> In response, we have added new experiments and comparisons for the iterative LQ method in Section B.5 of the revised manuscript. Our EMP has the lower collision rates than iterative LQ across all three case studies: 0.0\% (EMP) vs. 50.0\% (iterative LQ) in Case 1, 0.0\% (EMP) vs. 96.8\% (iterative LQ) in Case 2, 0.5\% (EMP) vs. 70.8\% (iterative LQ) in Case 3. We hypothesize that iterative LQ performs poorly in these settings because it is less capable of handling the non-smooth value functions caused by large collision penalties.
>
> **Q4. How sensitive is the proposed framework to hyperparameter tuning?**
>
> In response, we have added new experiments, comparisons, and discussions regarding the weight parameters in the loss function (Eq. 14), the choice of fine-grid size $h$, and the effect of dynamic noise. These additions are included in Section B.5 of the revised manuscript. EMP with a balanced weight parameter achieves the best safety performance, suggesting that proper weighting is crucial to improve the accuracy of value approximations in our framework. This observation is consistent with theoretical insights reported in prior studies [7, 8]. Moreover, EMP with small fine-grid sizes ($h=0.01\ \mathrm{s}$ and $h=0.02\ \mathrm{s}$) attains high safety performance across all three case studies. This aligns with the role of multigrid refinement: by enforcing the dynamic programming principle in Eq. 4 along sampled rollouts, small fine-grid resolutions lead to more accurate value approximations and, consequently, safer control policies. Finally, EMP with low dynamic noise maintains good performance in the 5D and 9D settings but fails in the 13D case. This outcome is expected: EMP is designed for deterministic dynamics rather than stochastic systems. The 13D scenario features highly nonlinear dynamics and a high-dimensional state space, where injected noise amplifies approximation errors in the learned value function and increases the sensitivity of the resulting control policy to rollout deviations.

---

> ### Author Response · Authors · 2025-11-25
> **Response to Reviewer MJVr (3/3)**
>
> **Reference**
>
> [1] Altarovici A, Bokanowski O, Zidani H. A general Hamilton-Jacobi framework for non-linear state-constrained control problems[J]. ESAIM: Control, Optimisation and Calculus of Variations, 2013.
>
> [2] Lee D, Tomlin C J. Hamilton-Jacobi equations for two classes of state-constrained zero-sum games[J]. arXiv preprint arXiv:2106.15006, 2021.
>
> [3] Zhang L, Ghimire M, Zhang W, et al. Value approximation for two-player general-sum differential games with state constraints[J]. IEEE Transactions on Robotics, 2024.
>
> [4] Bui M, Giovanis G, Chen M, et al. Optimizeddp: An efficient, user-friendly library for optimal control and dynamic programming[J]. arXiv preprint arXiv:2204.05520, 2022.
>
> [5] Nakamura-Zimmerer T, Gong Q, Kang W. Adaptive deep learning for high-dimensional Hamilton--Jacobi--Bellman equations[J]. SIAM Journal on Scientific Computing, 2021.
>
> [6] Betts J T. Survey of numerical methods for trajectory optimization[J]. Journal of guidance, control, and dynamics, 1998.
>
> [7] Wang S, Teng Y, Perdikaris P. Understanding and mitigating gradient flow pathologies in physics-informed neural networks[J]. SIAM Journal on Scientific Computing, 2021.
>
> [8] Wang S, Yu X, Perdikaris P. When and why PINNs fail to train: A neural tangent kernel perspective[J]. Journal of Computational Physics, 2022.

---

### Author Response · Authors · 2025-11-30
**Summary of the Rebuttal**

We sincerely thank all reviewers and the AC for reviewing our manuscript and rebuttal. In our rebuttal, we address all reviewer concerns point by point, revise the manuscript with changes marked in blue, and add new experiments and discussions in Section B.5 and B.6 of the revised manuscript.

**Summary of the paper**

Our work proposes a fully self-supervised framework for deriving safe control policies, addressing a key limitation of prior studies, e.g., [1], which rely on supervised values or gradients. We introduce a self-supervised value–gradient dynamics module to learn value functions for solving two-player general-sum games and validate its effectiveness across three case studies. In addition, Multigrid FAS has not yet been applied to HJ-based PINN training, especially in the context of epigraph technique and general-sum differential games, and our method fills this gap. We evaluate our EMP across three cases with both linear and nonlinear dynamics. Experimental results show that EMP achieves higher safety performance than existing epigraph-based baselines (EPPO, EHPINN, EPIML) and yields closed-loop trajectories that closely match ground-truth solutions.

Some common questions emerge in all the comments. We summarize these questions and our clarifications below.

**(1) Non-epigraph baselines.**

We initially compare our method against existing epigraph-based baselines. However, both Reviewer MJVr and igoi suggest comparing EMP with non-epigraph baselines: iterative LQ and MADDPG. In response, we implement the additional experiments and provide the results and discussion in Section B.5. EMP consistently has higher safety performance than both iterative LQ and MADDPG, as shown in Fig. 33-38.

**(2) Fine-grid size $h$ choice.**

Based on the suggestions of Reviewer MJVr and eCz1, we evaluate the effect of the multigrid timestep $h$ and provide additional experiments using three different $h$ settings in Section B.5. EMP with small fine-grid sizes ($h=0.01\ \mathrm{s}$ and $h=0.02\ \mathrm{s}$) attains high safety performance, as shown in Fig. 21-24.

**(3) Weight parameters in loss function.**

Reviewer MJVr and eCz1 suggest evaluating the effect of weight parameter tuning. In response, we implement additional experiments using three distinct weight configurations, and we report the corresponding results and discussion in Section B.5. EMP with a balanced weight parameter achieves the best safety performance (see Fig. 25-28), which is consistent with theoretical insights reported in prior studies [2, 3].

**(4) Dynamic noise.**

Following the suggestions of Reviewer MJVr and igoi, we add three dynamic noise levels (low, medium, and high) to evaluate our model. The experimental results and discussion are in Section B.5 (see Fig. 29-32). EMP with low dynamic noise maintains good safety performance in 5D and 9D case studies.

**(5) Equilibrium verification of learned model.**

We have evaluated the learned model by comparing its generated closed-loop trajectories with ground-truth PMP solutions (see Fig. 4, 6, 10, 13, 16, and 19) and by comparing the backward reachable safe sets with two ground-truth methods (see Fig. 20). In response to questions from Reviewer MJVr and igoi, we additionally include equilibrium verification results in Table 3 (Section B.5), providing further evidence that the learned model approximates the Nash equilibrium in both values and policies.

**(6) Explanation of epigraph technique.**

Following Reviewer iVJj's comments, we add an intuitive illustration and a step-by-step toy example in Section B.6 to explain the epigraph technique and its associated HJ PDEs.

We would like to again thank the AC for their considerable efforts in ensuring a fair and thorough review process. We hope that our summary helps the AC make an informed final decision.

Sincerely,

The Authors

[1] Zhang L, Ghimire M, Zhang W, et al. Value approximation for two-player general-sum differential games with state constraints[J]. IEEE Transactions on Robotics, 2024.

[2] Wang S, Teng Y, Perdikaris P. Understanding and mitigating gradient flow pathologies in physics-informed neural networks[J]. SIAM Journal on Scientific Computing, 2021.

[3] Wang S, Yu X, Perdikaris P. When and why PINNs fail to train: A neural tangent kernel perspective[J]. Journal of Computational Physics, 2022.

---

### Meta-Review · Area_Chair_NphK · 2026-01-02

**Summary:**

The paper presents a self-supervised framework combining epigraph-based PINNs, value/gradient rollouts, and multigrid refinement to solve two-player general-sum differential games with state constraints and demonstrates strong empirical safety performance across several benchmarks. Reviewers found the approach technically sound and the experiments thorough, but raised concerns that ultimately outweigh the strengths. In particular, questions remained about the novelty of the core contributions relative to closely related prior and concurrent work, the lack of strong theoretical guarantees, and the limited strength of equilibrium verification beyond indirect empirical evidence. These issues informed the recommendation to reject.

**Reviewer Concerns:**

The rebuttal addressed several important points, including adding non-epigraph baselines, expanding hyperparameter and robustness analyses, improving clarity and exposition, and providing additional evaluations of equilibrium-related metrics. However, key concerns remain outstanding. The contribution is still perceived as primarily an integration of existing ideas rather than a clearly differentiated methodological advance, and the theoretical analysis does not establish convergence to a Nash equilibrium. While equilibrium verification was strengthened, it remains indirect and application-specific. Additionally, concerns about overlap with prior or concurrent work raised by a reviewer warrant caution in evaluating the paper’s novelty and positioning. Taken together, these unresolved issues limit the case for acceptance despite strong empirical results.

**Reviewer Scores:**

Reviewer MJVr would likely remain around the borderline threshold, potentially increasing their score. Reviewer igoi would likely remain at 6. Reviewer iVJj would likely remain at 6. Reviewer eCz1 would likely remain around the borderline level, with technical improvements offset by continued concerns regarding novelty and overlap.

---

### Decision · Program_Chairs · 2026-01-26

Reject